**SOFTWARE**

# Microbiome meta-analysis and cross-disease comparison enabled by the SIAMCAT machine learning toolbox

Jakob Wirbel[1] , Konrad Zych[1,2] , Morgan Essex[1,3] , Nicolai Karcher[1,4] , Ece Kartal[1] , Guillem Salazar[5] , Peer Bork[1,6,7,8] , Shinichi Sunagawa[5] and Georg Zeller[1*] 

* Correspondence: zeller@embl.de
[1]Structural and Computational Biology Unit, European Molecular Biology Laboratory (EMBL), 69117 Heidelberg, Germany
Full list of author information is available at the end of the article

## Abstract

The human microbiome is increasingly mined for diagnostic and therapeutic biomarkers using machine learning (ML). However, metagenomics-specific software is scarce, and overoptimistic evaluation and limited cross-study generalization are prevailing issues. To address these, we developed SIAMCAT, a versatile R toolbox for ML-based comparative metagenomics. We demonstrate its capabilities in a meta-analysis of fecal metagenomic studies (10,803 samples). When naively transferred across studies, ML models lost accuracy and disease specificity, which could however be resolved by a novel training set augmentation strategy. This reveals some biomarkers to be disease-specific, with others shared across multiple conditions. SIAMCAT is freely available from siamcat.embl.de.

**Keywords:** Microbiome data analysis, Machine learning, Statistical modeling, Microbiome-wide association studies (MWAS), Meta-analysis

## Introduction

The study of microbial communities through metagenomic sequencing has begun to uncover how communities are shaped by—and interact with—their environment, including the host organism in the case of gut microbes [1, 2]. Especially within a disease context, differences in human gut microbiome compositions have been linked to many common disorders, for example, colorectal cancer [3], inflammatory bowel disease [4, 5], or arthritis [6, 7]. As the microbiome is increasingly recognized as an important factor in health and disease, many possibilities for clinical applications are emerging for diagnosis [8, 9], prognosis, or prevention of disease [10].

The prospect of clinical applications also comes with an urgent need for methodological rigor in microbiome analyses in order to ensure the robustness of findings. It is necessary to assess the clinical value of biomarkers identified from the microbiome in an unbiased manner—not only by their statistical significance, but more importantly also by their prediction accuracy on independent samples (allowing for external

validation). Machine learning (ML) models—ideally interpretable and parsimonious ones—are crucial tools to identify and validate such microbiome signatures. Setting up ML workflows however poses difficulties for novices. In general, it is challenging to assess their performance in an unbiased way, to apply them in cross-study comparisons, and to avoid confounding factors, for example, when disease and treatment effects are intertwined [11]. For microbiome studies, additional issues arise from key characteristics of metagenomic data such as large technical and inter-individual variation [12], experimental bias [13], compositionality of relative abundances, zero inflation, and non-Gaussian distribution, all of which necessitate data normalization in order for ML algorithms to work well.

While several statistical analysis tools have been developed specifically for microbiome data, they are generally limited to testing for differential abundance of microbial taxa between groups of samples and do not allow users to evaluate their predictivity as they do not comprise full ML workflows for biomarker discovery [14–16]. To overcome the limitations of testing-based approaches, several researchers have explicitly built ML classifiers to distinguish case and control samples [17–24]; however, the software resulting from these studies is generally not easily modified or transferred to other classification tasks or data types. To our knowledge, a powerful yet user-friendly computational ML toolkit tailored to the characteristics of microbiome data has not yet been published.

Here, we present SIAMCAT (Statistical Inference of Associations between Microbial Communities And host phenoTypes), a comprehensive toolbox for comparative metagenome analysis using ML, statistical modeling, and advanced visualization approaches. It also includes functionality to identify and visually explore confounding factors. To demonstrate its versatile applications, we conducted a large-scale ML meta-analysis of 130 classification tasks from 50 gut metagenomic studies (see Table 1) that have been processed with a diverse set of taxonomic and functional profiling tools. Based on this large-scale application, we arrive at recommendations for sensible parameter choices for the ML algorithms and preprocessing strategies provided in SIAMCAT. Moreover, we illustrate how several common pitfalls of ML applications can be avoided using the statistically rigorous approaches implemented in SIAMCAT. When considering the cross-study application of ML models, we note prevailing problems with type I error control (i.e., elevated false-positive rate, abbreviated as FPR) as well as disease specificity for ML models naively transferred across datasets. To alleviate these issues, we propose a strategy based on sampling additional external controls during cross-validation (which we call control augmentation). This enables cross-disease comparison of gut microbial biomarkers. Lastly, we showcase how SIAMCAT facilitates meta-analyses in an application to fecal shotgun metagenomic data from five independent studies of Crohn's disease. SIAMCAT is implemented in the R programming language and freely available from siamcat.embl.de or Bioconductor.

## Results

### Machine learning and statistical analysis workflows implemented in SIAMCAT

The SIAMCAT R package is a versatile toolbox for analyzing microbiome data from case-control studies. The default workflows abstract from and combine many of the

**Table 1** Overview of diseases and datasets included in the ML meta-analysis

| Disease | Disease abbr. | Datasets | Data type |
|---|---|---|---|
| Ankylosing spondylitis | AS | [7] | Shotgun |
| Rheumatoid arthritis | ART | [25] | 16S rRNA |
|  |  | [6] | Shotgun |
| Type 1 diabetes | T1D | [26] | 16S rRNA |
| Crohn's disease | CD | [5, 27–30] | Shotgun |
| Ulcerative colitis | UC | [5, 27, 30] | Shotgun |
| Inflammatory bowel disease | IBD | [4, 31–33] | 16S rRNA |
| Colorectal cancer | CRC | [8, 34–39] | Shotgun |
|  |  | [8, 40–42] | 16S rRNA |
| Advanced colorectal adenoma(s) | ADA | [8, 34, 38, 39] | Shotgun |
| Atherosclerotic cardiovascular disease | ACVD | [43] | Shotgun |
| Hypertension Pre-hypertension | HT pHT | [44] | Shotgun |
| *Clostridioides difficile* infection | CDI | [45, 46] | 16S rRNA |
| Enteric diarrheal disease | EDD | [47] | 16S rRNA |
| HIV infection | HIV | [48–50] | 16S rRNA |
| Liver cirrhosis | LIV | [51] | Shotgun |
|  |  | [52] | 16S rRNA |
| Non-alcoholic fatty liver disease | NAFLD | [53, 54] | Shotgun |
|  |  | [55, 56] | 16S rRNA |
| Parkinson's disease | PAR | [57] | Shotgun |
|  |  | [58] | 16S rRNA |
| Autism spectrum disorder | ASD | [59, 60] | 16S rRNA |
| Obesity | OB | [61] | Shotgun |
|  |  | [62–65] | 16S rRNA |
| Metabolic syndrome | MS | [66] | Shotgun |
| Type 2 diabetes | T2D | [67, 68] | Shotgun |
| Impaired glucose tolerance | IGT | [67] | Shotgun |

complex steps that these workflows entail and that can be difficult to implement correctly for non-experts. To increase ease of use, SIAMCAT interfaces with the popular phyloseq package [69], and design and parameter choices are carefully adapted to metagenomic data analysis. In addition to functions for statistical testing of associations, SIAMCAT workflows include ML procedures, also encompassing data preprocessing, model fitting, performance evaluation, and visualization of the results and models (Fig. 1a). Core ML functionality is based on the mlr package [70]. The input for SIAMCAT consists of a feature matrix (abundances of microbial taxa, genes, or pathways across all samples), a group label (case-control information for all samples), and optional meta-variables (such as demographics, lifestyle, and clinical records of sample donors or technical parameters of data acquisition).

To demonstrate the main workflow and primary outputs of the SIAMCAT package (see the "Methods" section and SIAMCAT vignettes), we analyzed a representative dataset [27] consisting of 128 fecal metagenomes from patients with ulcerative colitis (UC) and non-UC controls (Fig. 1). UC is a subtype of inflammatory bowel disease

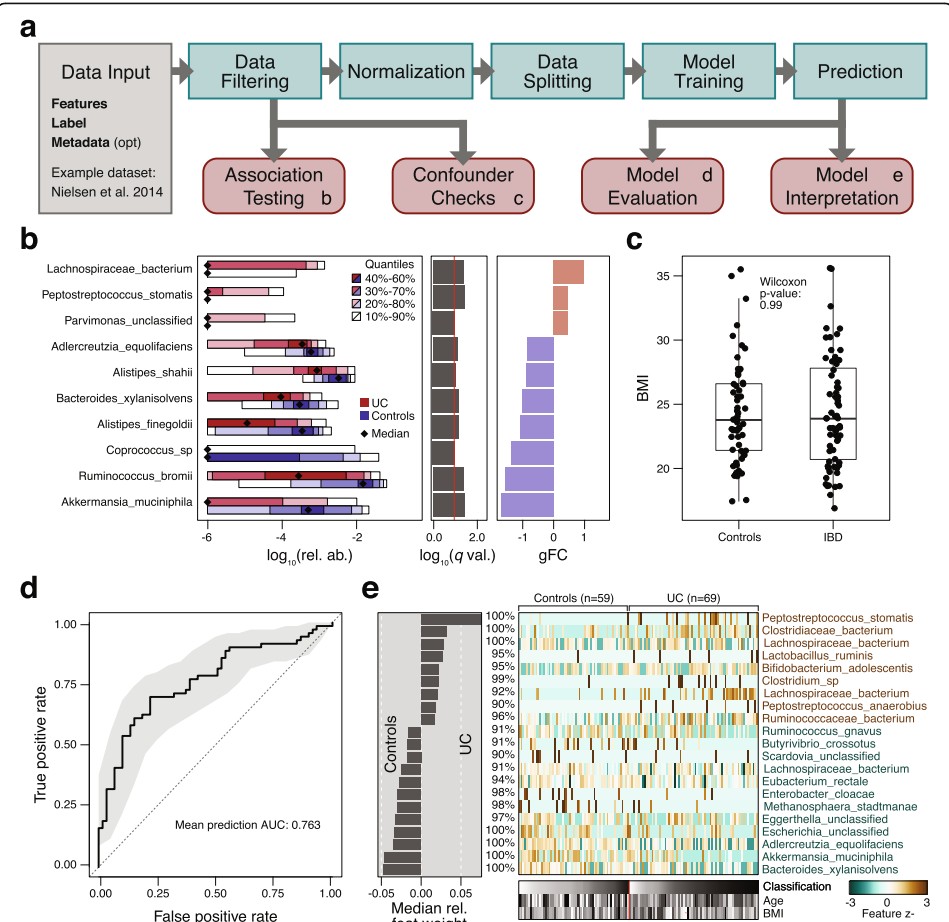

**Fig. 1** SIAMCAT statistical and machine learning approach model differences between the groups of microbiome samples. **a** Each step in the SIAMCAT workflow (green boxes) is implemented by a function in the R/Bioconductor package (see SIAMCAT vignettes). Functions producing graphical output (red boxes) are illustrated in **b**–**e** for an exemplary analysis using a dataset from Nielsen et al. [27] which contains ulcerative colitis (UC) patients and non-UC controls. **b** Visualization of the univariate association testing results. The left panel visualizes the distributions of microbial abundance data differing significantly between the groups. Significance (after multiple testing correction) is displayed in the middle panel as horizontal bars. The right panel shows the generalized fold change as a non-parametric measure of effect size [37]. **c** SIAMCAT offers statistical tests and diagnostic visualizations to identify potential confounders by testing for associations between such meta-variables as covariates and the disease label. The example shows a comparison of body mass index (BMI) between the study groups. The similar distributions between cases and controls suggest that BMI is unlikely to confound UC associations in this dataset. Boxes denote the IQR across all values with the median as a thick black line and the whiskers extending up to the most extreme points within 1.5-fold IQR. **d** The model evaluation function displays the cross-validation error as a receiver operating characteristic (ROC) curve, with a 95% confidence interval shaded in gray and the area under the receiver operating characteristic curve (AUROC) given below the curve. **e** SIAMCAT finally generates visualizations aiming to facilitate the interpretation of the machine learning models and their classification performance. This includes a barplot of feature importance (in the case of penalized logistic regression models, bar width corresponds to coefficient values) for the features that are included in the majority of models fitted during cross-validation (percentages indicate the respective fraction of models containing a feature). A heatmap displays their normalized values across all samples (as used for model fitting) along with the classification result (test predictions) and user-defined meta-variables (bottom)

(IBD), a chronic inflammatory condition of the gastrointestinal tract that has been associated with dramatic changes in the gut microbiome [5, 71]. As input, we used species-level taxonomic profiles available through the *curatedMetagenomicsData* R package [72].

After data preprocessing (unsupervised abundance and prevalence filtering, Fig. 1a and the "Methods" section), univariate associations of single species with the disease are computed using the non-parametric Wilcoxon test (which has been shown for metagenomic data to reliably control the false discovery rate in contrast to many other tests proposed [73]), and the results are visualized (using the *check.associations* function). The association plot displays the distribution of microbial relative abundance, the significance of the association, and a generalized fold change as a non-parametric measure of effect size [37] (Fig. 1b).

The central component of SIAMCAT consists of ML procedures, which include a selection of normalization methods (*normalize.features*), functionality to set up a cross-validation scheme (*create.data.split*), and interfaces to different ML algorithms, such as LASSO, Elastic Net, and random forest (offered by the mlr package [70]) [74–76]. As part of the cross-validation procedure, models can be trained (*train.model*) and applied to make predictions (*make.predictions*) on samples not used for training. Based on these predictions, the performance of the model is assessed (*evaluate.predictions*) using the area under the receiver operating characteristic (ROC) curve (AUROC) (Fig. 1d). SIAMCAT also provides diagnostic plots for the interpretation of ML models (*model.interpretation.plot*) which display the importance of individual features in the classification model, normalized feature distributions as heatmaps, next to sample meta-variables (optionally, see Fig. 1c, e).

Expert users can readily customize and flexibly recombine the individual steps in the described workflow above. For example, filtering and normalization functions can be combined or omitted before ML models are trained or association statistics calculated. To demonstrate its versatility beyond the workflow presented in Fig. 1a, we used SIAMCAT to reproduce two recent ML meta-analyses of metagenomic datasets [19, 20]. By implementing the same workflows as described in the respective papers, we could generate models with very similar accuracy (within the 95% confidence interval) for all datasets analyzed (Additional file 1: Figure S1).

### Confounder analysis using SIAMCAT

As many biological and technical factors beyond the primary phenotype of interest can influence microbiome composition [1], microbiome association studies are often at a high risk of confounding, which can lead to spurious results [11, 77–79]. To minimize this risk, SIAMCAT provides a function to optionally examine potential confounders among the provided meta-variables. In the example dataset from [27], control samples were obtained from both Spanish and Danish subjects, while UC samples were only taken from Spanish individuals (Fig. 2a). Here, the meta-variable "country" could be viewed as a surrogate variable for other (often difficult-to-measure) factors, which can influence microbiome composition, such as diet, lifestyle, or technical differences between studies. The strong association of the "country" meta-variable with the disease status (SIAM CAT computes such associations using Fisher's exact test or the Wilcoxon test for discrete and continuous meta-variables, respectively; see Fig. 2a) hints at the possibility that

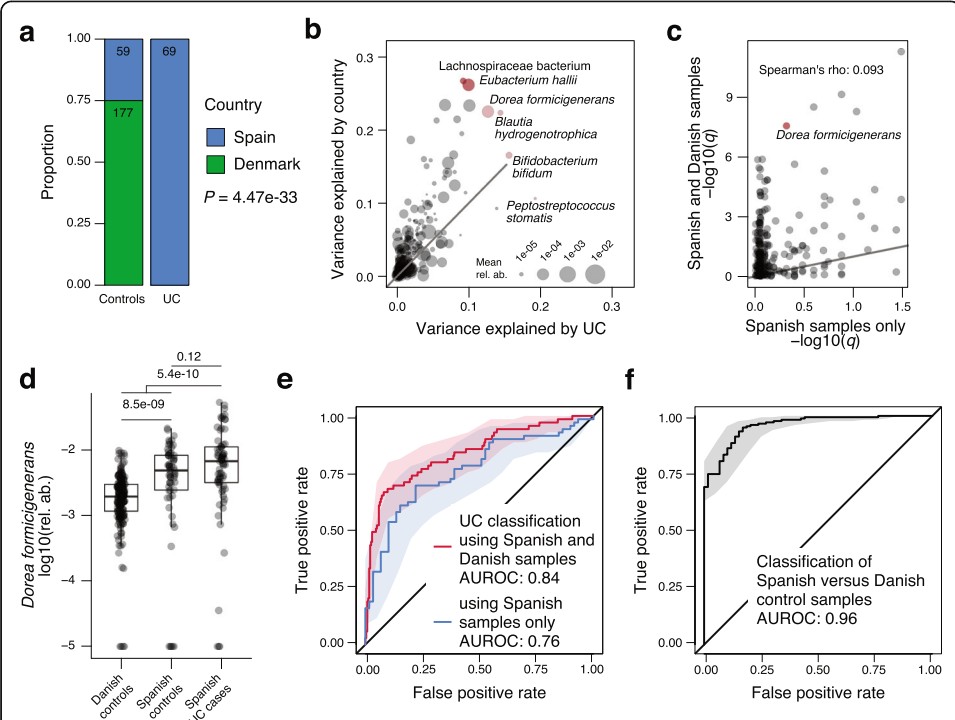

**Fig. 2** Analysis of covariates that potentially confound microbiome-disease associations and classification models. The UC dataset from Nielsen et al. [27] contains fecal metagenomes from subjects enrolled in two different countries and generated using different experimental protocols (data shown is from *curatedMetagenomicData* with CD cases and additional samples per subject removed). **a** Visualizations from the SIAMCAT confounder checks reveals that only control samples were taken from Denmark suggesting that any (biological or technical) differences between Danish and Spanish samples might confound a naive analysis for UC-associated differences in microbial abundances. **b** Analysis of variance (using ranked abundance data) shows many species differ more by country than by disease, with several extreme cases highlighted. **c** When comparing (FDR-corrected) *P* values obtained from SIAMCAT's association testing function applied to the whole dataset (*y*-axis) to those obtained for just the Danish samples (*x*-axis), only a very weak correlation is seen and strong confounding becomes apparent for several species including *Dorea formicigenerans* (highlighted). **d** Relative abundance differences for *Dorea formicigenerans* are significantly larger between countries than between Spanish UC cases and controls (*P* values from Wilcoxon test) (see Fig. 1c for the definition of boxplots). **e** Distinguishing UC patients from controls with the same workflow is possible with lower accuracy when only samples from Spain are used compared to the full dataset containing Danish and Spanish controls. This implies that in the latter case, the machine learning model is confounded as it exploits the (stronger) country differences (see **c** and **f**), not only UC-associated microbiome changes. **f** This is confirmed by the result that control samples from Denmark and Spain can be very accurately distinguished with an AUROC of 0.96 (using SIAMCAT classification workflows)

associations computed with the full dataset could be confounded by the country of the sample donor.

To quantify this confounding effect on individual microbial features, SIAMCAT additionally provides a plot for each meta-variable that shows the variance explained by the label in comparison with the variance explained by the meta-variable for each individual feature (Fig. 2b, implemented in the *check.confounder* function). In our example case, several microbial species are strongly associated with both the disease phenotype (UC vs control) and the country, indicating that their association with the label might simply be an effect of technical and/or biological differences between samples taken and data processed in the different countries.

To further investigate this confounder, we used SIAMCAT to compute statistical association for the full dataset (including the Danish control samples) and the reduced dataset containing only samples from Spanish individuals (using the *check.association* function). The finding that $P$ values were uncorrelated between the two datasets (Fig. 2c) directly quantified the effect of confounding by country on the disease-association statistic. The potential severity of this problem is highlighted by a comparison of the relative abundance of *Dorea formicigenerans* across subjects: the differences between UC cases and controls are only significant when Danish control samples are included, but not when restricted to Spanish samples only (Fig. 2d), exemplifying how confounders can lead to spurious associations.

Finally, confounding factors can not only bias statistical association tests, but can also impact the performance of ML models. A model trained to distinguish UC patients from controls seemingly performs better if the Danish samples are included (AUROC of 0.84 compared to 0.76 if only using Spanish samples), because the differences between controls and UC samples are artificially inflated by the differences between Danish and Spanish samples (Fig. 2e). How these overall differences between samples taken in different countries can be exploited by ML models can also be directly quantified using SIAMCAT workflows. The resulting model trained to distinguish between control samples from the two countries can do so with almost perfect accuracy (AUROC of 0.96) (Fig. 2f). This analysis demonstrates how confounding factors can lead to exaggerated performance estimates for ML models.

In summary, SIAMCAT can help to detect influential confounding factors that have the potential to bias statistical associations and ML model evaluations (see Additional file 1: Figure S2 for additional examples).

### Advanced machine learning workflows

When designing more complex ML workflows involving feature selection steps or applications to time series data, it becomes more challenging to set up cross-validation procedures correctly. Specifically, it is important to estimate how well a trained model would generalize to an independent test set, which is typically the main objective of microbial biomarker discovery. An incorrect ML procedure, in which information leaks from the test to the training set, can result in overly optimistic (i.e., overfitted) performance estimates. Two pitfalls that can lead to overfitting and poor generalization to other datasets (Fig. 3a) are frequently encountered in ML analyses of microbiome and other biological data, even though the issues are well described in the statistics literature [80–82]. These issues, namely supervised feature filtering and naive splitting of dependent samples, can be exposed by testing model performance in an external validation set, which has not been used during cross-validation at all (Fig. 3b).

The first issue arises when feature selection taking label information into account (supervised feature selection) is naively combined with subsequent cross-validation on the same data [81]. This incorrect procedure selects features that are associated with the label (e.g., by testing for differential abundance) on the complete dataset leaving no data aside for an unbiased test error estimation of the whole ML procedure. To avoid overfitting, correct supervised feature selection should always be nested into cross-validation (that is, the supervised feature selection has to be applied to each training

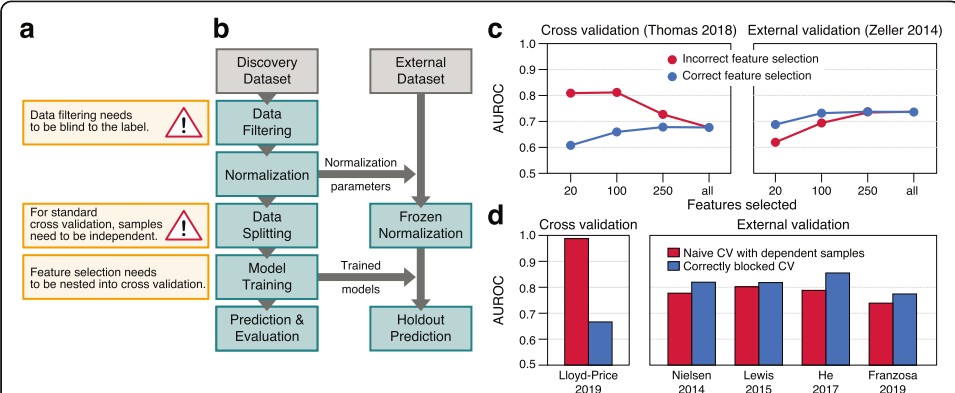

**Fig. 3** SIAMCAT aids in avoiding common pitfalls leading to a poor generalization of machine learning models. **a** Incorrectly setup machine learning workflows can lead to overoptimistic accuracy estimates (overfitting): the first issue arises from a naive combination of feature selection on the whole dataset and subsequent cross-validation on the very same data [80]. The second one arises when samples that were not taken independently (as is the case for replicates or samples taken at multiple time points from the same subject) are randomly partitioned in cross-validation with the aim to assess the cross-subject generalization error (see the main text). **b** External validation, for which SIAMCAT offers analysis workflows, can expose these issues. The individual steps in the workflow diagram correspond to SIAMCAT functions for fitting a machine learning model and applying it to an external dataset to assess its external validation accuracy (see SIAMCAT vignette: holdout testing with SIAMCAT). **c** External validation shows overfitting to occur when feature selection and cross-validation are combined incorrectly in a sequential manner, rather than correctly in a nested approach. The correct approach is characterized by a lower (but unbiased) cross-validation accuracy, but better generalization accuracy to external datasets (see header for datasets used). The fewer features are selected, the more pronounced the issue becomes, and in the other extreme case ("all"), feature selection is effectively switched off. **d** When dependent observations (here by sampling the same individuals at multiple time points) are randomly assigned to cross-validation partitions, effectively the ability of the model to generalize across time points, but not across subjects, is assessed. To correctly estimate the generalization accuracy across subjects, repeated measurements need to be blocked, all of them either into the training or test set. Again, the correct procedure shows lower cross-validation accuracy, but higher external validation accuracy

fold of the cross-validation separately). To illustrate the extent of overfitting resulting from the incorrect approach, we used two datasets of colorectal cancer (CRC) patients and controls and performed both the incorrect and correct ways of supervised feature selection. As expected, the incorrect feature selection led to inflated performance estimates in cross-validation but lower generalization to an external dataset, whereas the correct procedure gave a better estimate of the performance in the external test set; the fewer features were selected, the more the performance in the external datasets dropped (see Fig. 3c). SIAMCAT readily provides implementations of the correct procedure and additionally takes care that the feature filtering and normalization of the whole dataset are blind to the label (therefore called unsupervised), thereby preventing accidental implementation of the incorrect procedure.

The second issue tends to occur when samples are not independent [82]. For example, microbiome samples taken from the same individual at different time points are usually a lot more similar to each other than those from different individuals (see [12] and Additional file 1: Figure S3). If these dependent samples are randomly split in a standard cross-validation procedure, so that some could end up in the training set and others in the test set, it is effectively estimated how well the model generalizes across time points (from the same individual) rather than across individuals. To avoid this, dependent measurements need to be blocked during cross-validation, ensuring that

measurements of the same individual are assigned to the same test set. How much the naive procedure can overestimate the performance in cross-validation and underperform in external validation compared to the correctly blocked procedure is demonstrated here using the iHMP dataset, which contains several samples per subject [30]. Although the cross-validation accuracy appears dramatically lower in the correct compared to the naive procedure, generalization to other datasets of the same disease is higher with the correctly blocked model (Fig. 3d). SIAMCAT offers the possibility to block the cross-validation according to meta-variables by simply providing an additional argument to the respective function call (see also SIAMCAT vignettes).

### Large-scale machine learning meta-analysis

Previous studies that applied ML to microbiome data [17–20] have compared and discussed the performance of several learning algorithms. However, their recommendations were based on the analysis of a small number of datasets which were technically relatively homogeneous. To overcome this limitation and to demonstrate that SIAMCAT can readily be applied to various types of input data, we performed a large-scale ML meta-analysis of case-control gut metagenomic datasets. We included taxonomic profiles obtained with the RDP taxonomic classifier [83] for 26 datasets based on 16S rRNA gene sequencing [20]; additionally, taxonomic profiles generated from 12 and 24 shotgun metagenomic datasets using either MetaPhlAn2 [84] or mOTUs2 [85], respectively, as well as functional profiles obtained with HUMAnN2 [86] or with eggNOG 4.5 [87] for the same set of shotgun metagenomic data were included (in total 130 classification tasks, see Table 1 and Additional file 2: Table S1 for information about included datasets).

Focusing first on intra-study results, we found that given a sufficiently large input dataset (with at least 100 samples), SIAMCAT models are generally able to distinguish reasonably well between cases and controls: the majority (58%) of these datasets in our analysis could be classified with an AUROC of 0.75 or higher—compared to only 36% of datasets with fewer than 100 samples (Fig. 4a–c, Additional file 1: Figures S4 and S5 and the "Methods" section). Of note, accurate ML-based classification was possible even for datasets in which cases and controls could not easily be separated using beta-diversity analyses (Additional file 1: Figure S6), indicating that a lack of separation in ordination analysis does not preclude ML-based workflows to extract accurate microbiome signatures. In the datasets for which a direct comparison of mOTUs2 and MetaPhlAn2 profiles was possible, we did not find any consistent trend towards either profiling method (paired Wilcoxon $P = 0.41$, see Additional file 1: Figure S7). When comparing taxonomic and functional profiles derived from the same dataset, we found a high correlation between AUROC values (Pearson's $r = 0.92$, $P < 2 \times 10^{-16}$), although on average taxonomic profiles performed slightly better than functional profiles (Additional file 1: Figure S7). Taken together, this indicates that SIAMCAT can extract accurate microbiome signatures (model cross-validation AUROC $> 0.75$ in 64 of 130 classification tasks) from a range of different input profiles commonly used in microbiome research.

SIAMCAT provides various methods for data filtering and normalization and interfaces to several ML algorithms through mlr [70]. This made it easy to explore the space

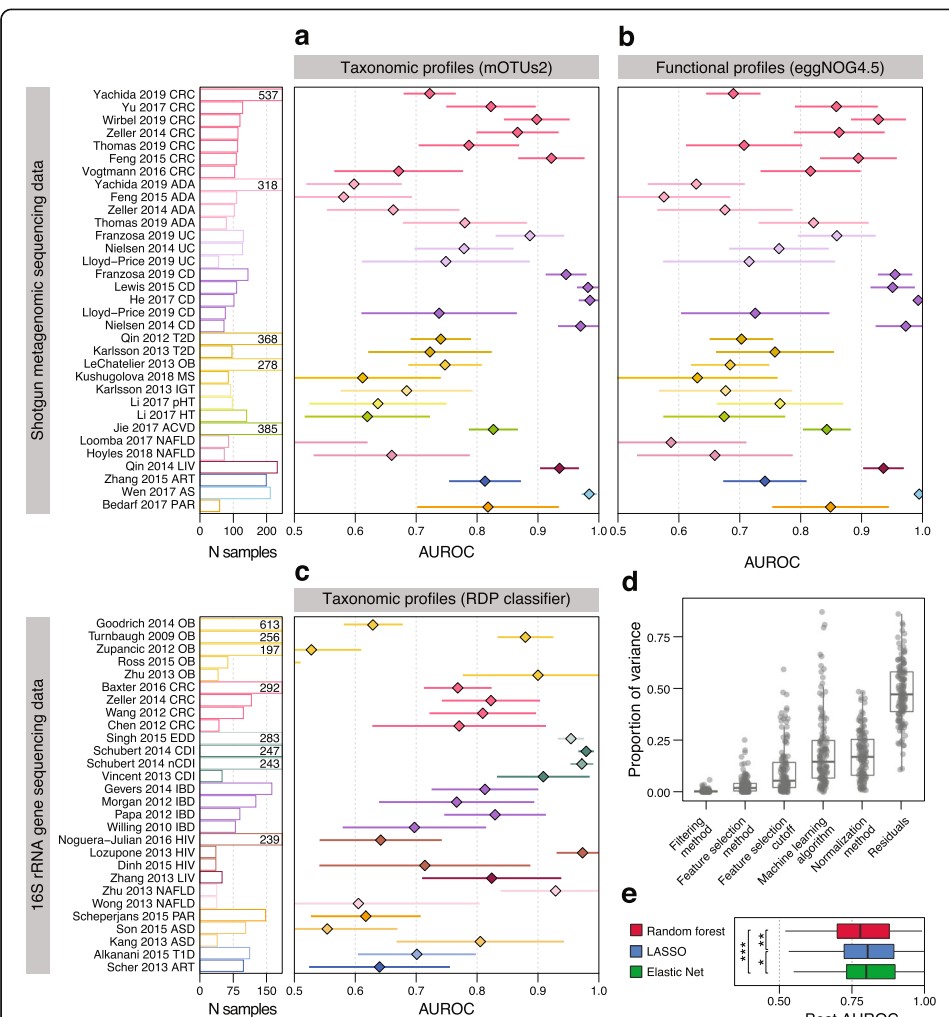

**Fig. 4** Large-scale application of the SIAMCAT machine learning workflow to human gut metagenomic disease association studies. **a** Application of SIAMCAT machine learning workflows to taxonomic profiles generated from fecal shotgun metagenomes using the mOTUs2 profiler. Cross-validation performance for discriminating between diseased patients and controls quantified by the area under the ROC curve (AUROC) is indicated by diamonds (95% confidence intervals denoted by horizontal lines) with sample size per dataset given as additional panel (cut at $N = 250$ and given by numbers instead) (see Table 1 and Additional file 2: Table S1 for information about the included datasets and key for disease abbreviations). **b** Application of SIAMCAT machine learning workflows to functional profiles generated with eggNOG 4.5 for the same datasets as in **a** (see Additional file 1: Figure S4, S7 for additional types of and comparison between taxonomic and functional input data). **c** Cross-validation accuracy of SIAMCAT machine learning workflows as applied to 16S rRNA gene amplicon data for human gut microbiome case-control studies [20] (see **a** for definitions). **d** Influence of different parameter choices on the resulting classification accuracy. After training a linear model to predict the AUROC values for each classification task, the variance explained by each parameter was assessed using an ANOVA (see the "Methods" section) (see Fig. 1 for the definition of boxplots). **e** Performance comparison of machine learning algorithms on gut microbial disease association studies. For each machine learning algorithm, the best AUROC values for each task are shown as boxplots (defined as in **d**). Generally, the choice of algorithm only has a small effect on classification accuracy, but both the Elastic Net and LASSO performance gains are statistically significant (paired Wilcoxon test: LASSO vs Elastic Net, $P = 0.001$; LASSO vs random forest, $P = 1e-08$; Elastic Net vs random forest, $P = 4e-14$)

of possible workflow configurations in order to arrive at recommendations about sensible default parameters. To test the influence of different parameter choices within the complete data analysis pipeline, we performed an ANOVA analysis to quantify their relative importance on the resulting classification accuracy (Fig. 4d and the "Methods" section). Whereas the choice of filtering method and feature selection regime has little influence on the results, the normalization method and ML algorithm explained more of the observed variance in classification accuracy. Analysis of the different normalization methods shows that most of the differences can be explained by a drop in performance for naively normalized data (only total sum scaling and no further normalization) in combination with LASSO or Elastic Net logistic regression (Additional file 1: Figure S8). In contrast, the random forest classifier depended much less on optimal data normalization. Lastly, we compared the best classification accuracy for each classification task across the different ML algorithms. Interestingly, in contrast to a previous report [19], this analysis indicates that on average Elastic Net logistic regression outperforms LASSO and random forest classifiers when considering the optimal choice of ML algorithm ($P = 0.001$ comparing Elastic Net to LASSO and $P = 4 \times 10^{-14}$ comparing it to random forest, Fig. 4e). In summary, this large-scale analysis demonstrates the versatility of the ML workflows provided by SIAMCAT and validates its default parameters as well as the robustness of classification accuracy to deviations from these.

Cross-study evaluation of microbiome signatures is crucial to establish their validity across patient populations. However, such comparisons are potentially hindered by inter-study differences in sample handling and data generation, with technical variation observed to often dominate over biological factors of interest [88–90]. Additionally, biological and clinical factors can contribute to inter-study differences. These not only include influences of geography, ethnicity, demographics, and lifestyle, but also how clinical phenotypes are defined and controls selected for each study [91].

Up to now, it has not been systematically explored how well microbiome-based ML models transfer across a range of diseases. To close this gap, we used our large-scale ML meta-analysis and trained ML models for each task using mOTUs2 taxonomic profiles as input (based on the previously established best-performing parameter set). We subsequently focused on models with reasonable cross-validation accuracy (AUROC > 0.75) and applied these to all remaining datasets to make predictions.

Cross-study application of ML models is straightforward within the same disease, since the model predictions on external datasets can easily be evaluated by an AUROC (Additional file 1: Figure S9, [37, 38]) under the assumption that case and control definitions are comparable between studies. However, when applying an ML model to a dataset from another disease, ROC analysis cannot be directly applied, since the cases the model was originally trained to detect are from another disease than those of the evaluation dataset. For this cross-disease application of ML models, we conducted extended evaluations, which specifically addressed the following two questions (see Additional file 1: Figure S10 and the "Methods" section). First, we asked to which extent the separation between cases and controls (in terms of prediction scores) would be maintained when control samples of a different study are used. We therefore employed a modified ROC analysis (comparing true-positive rates from cross-validation to external FPRs via AUROC) as a newly defined measure of cross-study portability of an ML

model. For convenience, we rescaled it to range between 0 (indicating a complete loss of discriminatory power on external data) and 1 (meaning that the ML model could be transferred to another dataset without loss of discrimination accuracy). Second, we asked how specific an ML model would be to the disease it was trained to recognize, or whether its FPR would be elevated when presented with cases from a distinct condition. This is of interest in the context of an ongoing debate on whether there is a general gut microbial dysbiosis or distinct compositional changes associated with each disease [19, 20, 92]. Disease-specific classifiers would also be of clinical relevance when applied to a general population: due to large differences in disease prevalence, a model for CRC (a condition with low prevalence) misclassifying many type 2 diabetes (T2D) patients (high prevalence) would in the general population detect many more (false) T2D cases than true CRC cases, and thus have very low precision. To quantify the prediction rate for other diseases of an ML model, i.e., its disease specificity, we assessed how many samples from a distinct disease would be mispredicted as positive for the disease the ML model was trained on at a cutoff adjusted to maintain a FPR of 10% on the cross-validation set.

These extended evaluations showed low cross-study portability on the majority of external datasets (apparent also from a more than twofold increase in false positives on average) for most models (Additional file 1: Figure S11). Similarly, (false-positive) predictions for other diseases were elevated for most models (by a factor of 2.8 on average), with the extreme case of the ankylosing spondylitis (AS) model predicting more than 90% of cases from other diseases to be AS positive (median across studies, Additional file 1: Figure S12). These evaluations indicate that naive ML model transfer is substantially impacted—if not rendered impossible—by biological and technical study heterogeneity, apparent from loss of general accuracy and disease specificity.

In order to improve the cross-study portability of ML models, we devised a strategy we call control augmentation, in which randomly selected control samples from independent microbiome population cohort studies [93–95] are added to the training set during model fitting (Fig. 5a, see the "Methods" section). This was motivated by the hypothesis that additional variability from a greater control pool comprising heterogeneous samples from multiple studies would enable classifiers to more specifically recognize disease signals while at the same time minimizing overfitting on peculiarities of a single dataset. However, a theoretical limitation of this approach is that the definition of controls can vary greatly across studies. In spite of this, in practice, we found control augmentation to greatly enhance cross-study portability uniformly across all ML models, both in cross-study analysis within the same condition and across different diseases (Fig. 5b, c, Additional file 1: Figure S9, S11). At the same time, cross-disease predictions decreased (Fig. 5c, d, Additional file 1: Figure S12) implying that it is an effective strategy to increase disease specificity of ML models.

The control augmentation strategy did not strongly depend on the set of controls used. We found large (> 250 samples) cohort studies to work well as a pool for control augmentation (allowing us to add five times the amount of control samples to each dataset). However, augmentation with fewer controls or with other datasets improved cross-study portability and disease specificity to almost the same effect (Additional file 1: Figure S13).

With cross-study portability greatly improved, we expect the remaining cross-disease predictions to be largely due to biological similarities between diseases rather than due to technical influences. In support of this, we show that CRC signatures have a tendency to cross-predict samples from patients with intestinal adenomas (ADA) or inflammatory bowel disease (CD), both of which are risk factors for CRC development [96]. Similarly, UC models cross-predict CD cases and vice versa, reflecting more general gut microbial changes, i.e., loss of beneficial commensal bacteria, that are shared across both types of inflammatory bowel disease [97]. In summary, we demonstrate that control augmentation is an effective strategy to broadly enable the validation of microbiome disease signatures across different studies, since it can overcome study-specific biases, which preclude the naive transfer of ML models.

When comparing microbiome signatures across diseases in more detail, we also revisited the question of whether microbiome alterations are specific to a disease, or signs of a general dysbiotic state [20]. As many ML algorithms, in particular (generalized) linear models, such as LASSO or Elastic Net logistic regression models, allow for model introspection, microbiome biomarkers can easily be extracted and their weight in the model directly quantified by (normalized) coefficient values. The model weights of the control-augmented models showed a clear clustering by disease in principal coordinate space revealing broad disease similarity patterns in terms of microbiome predictors that may reflect etiological similarities (Fig. 5e, not apparent from naively transferred ML models, Additional file 1: Figure S14). To obtain a more nuanced view of the gut bacterial taxa underlying these disease similarities, we analyzed individual mOTUs (grouped by genus membership) that were selected as predictors in disease models (Fig. 5f, to minimize bias from multiple studies of the same disease, we used the mean model for each disease and extracted those features whose weights accounted for more than 50% of the model, see the "Methods" section for details). We found some disease-enriched predictors to be very specific for a single disease, such as *Veillonella* spp. for LIV, *Bifidobacteria* and *Neisseria* mOTUs for AS, or *Gemella* and *Parvimonas* mOTUs for CRC. In contrast, species from other genera, for example, *Lactobacillus*, *Bacteroides*, or *Fusobacteria*, appear predictive of several diseases, although species and subspecies belonging to these vary in terms of their disease specificity (Additional file 1: Figure S15). Regarding control-enriched predictors, species from some genera are frequently depleted across multiple diseases (*Anaerostipes* and *Romboutisa*) while some diseases are marked by broad depletion of beneficial microbes, e.g., CD (consistent with [97]).

Overall, enabled by control augmentation as an effective strategy to improve cross-study portability of ML models, our cross-disease meta-analysis reveals both shared and disease-specific predictors as a basis for further development of microbiome-based diagnostic biomarkers.

### Meta-analysis of Crohn's disease gut microbiome studies

Microbiome disease associations being reported at an ever-increasing pace have also provided opportunities for comparisons across multiple studies of the same disease to assess the robustness of associations and the generalizability of ML models [19, 20, 37,

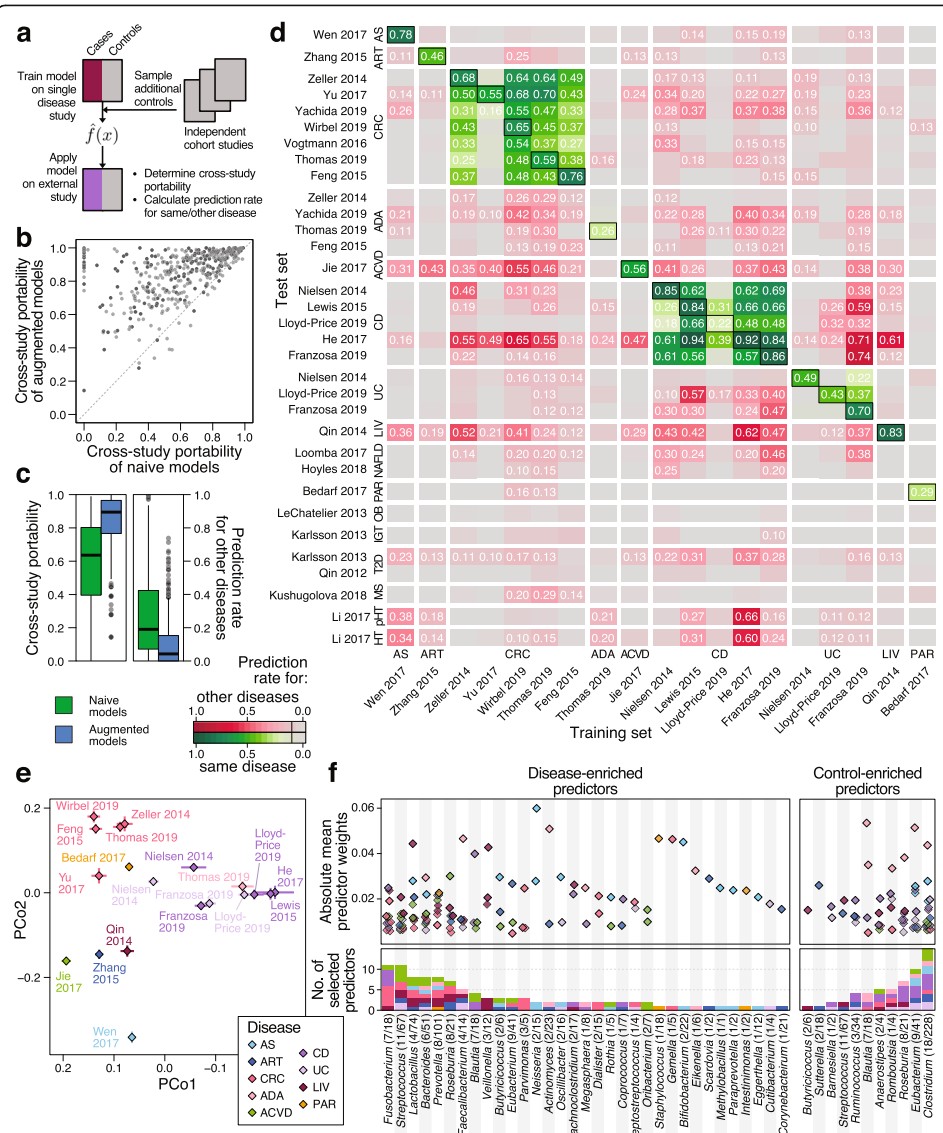

**Fig. 5** Control augmentation improves ML model disease specificity and reveals shared and distinct predictors. **a** Schematic of the control augmentation procedure: control samples from external cohort studies are added to the individual cross-validation folds during model training. Trained models are applied to external studies (either of a different or the same disease) to determine cross-study portability (defined as maintenance of type I error control on external control samples) and cross-disease predictions (i.e., false detection of samples from a different disease). **b** Cross-study portability was compared between naive and control-augmented models showing consistent improvements due to control augmentation. **c** Boxplots depicting cross-study portability (left) and prediction rate for other diseases (right) of naive and control-augmented models (see Fig. 1 for the definition of boxplots). **d** Heatmap showing prediction rates for other diseases (red color scheme) and for the same disease (green color scheme) for control-augmented models on all external datasets. True-positive rates of the models from cross-validation on the original study are indicated by boxes around the tile. Prediction rates over 10% are labeled. **e** Principal coordinate (PCo) analysis between models based on Canberra distance on model weights. Diamonds represent the mean per dataset in PCo space across cross-validation splits, and lines show the standard deviation. **f** Visualization of the main selected model weights (predictors corresponding to mOTUs, see the "Methods" section for the definition of cutoffs) by genus and disease. Absolute model weights are shown as a dot plot on top, grouped by genus (including only genera with unambiguous NCBI taxonomy annotation). Below, the number of selected weights per genus is shown as a bar graph, colored by disease (see **e** for color key). Genus labels at the bottom include the number of mOTUs with at least one selected weight followed by the number of mOTUs in the complete model weight matrix belonging to the respective genus

38]. To demonstrate SIAMCAT's utility in single-disease meta-analyses, we analyzed five metagenomic datasets [5, 27–30], all of which included samples from patients with Crohn's disease (CD) as well as controls not suffering from inflammatory bowel diseases (IBD). Raw sequencing data were consistently processed to obtain genus abundance profiles with mOTUs2 [85].

Based on SIAMCAT's *check.associations* function, we identified microbial genera that are significantly associated with CD in each study and visualized their agreement across studies (Fig. 6a, left panel). In line with previous findings [4], the gut microbiome of CD patients is characterized by a loss of diversity and many beneficial taxa. Though our re-analysis of the data from [30] could not identify any statistically significant genus-level associations, possibly due to the relatively small number of individuals or the choice of control samples obtained from patients with non-IBD gastrointestinal symptoms, the other four studies showed remarkable consistency among the taxa lost in CD patients, in particular, for members of the Clostridiales order.

We further investigated variation due to technical and biological differences between studies as a potential confounder using SIAMCAT's *check.confounder* function following a previously validated approach [37]. For many genera, variation can largely be attributed to heterogeneity among studies; the top five associated genera (cf. Figure 6a), however, vary much more with disease status, suggesting that their association with CD is only minimally confounded by differences between studies (Fig. 6b).

Next, we systematically assessed cross-study generalization of ML models trained to distinguish CD patients from controls using SIAMCAT workflows. To this end, we trained an Elastic Net model for each study independently and evaluated the performance of the trained models on the other datasets (Fig. 6c and the "Methods" section). Most models maintained very high classification accuracy when applied to the other datasets for external validation (AUROC > 0.9 in most cases); again with the exception of the model cross-validated on the data from [30], which exhibited substantially lower accuracy in both cross-validation and external validation.

We lastly assessed the importance of individual microbial predictors in the CD models. The LASSO, and to some extent also the Elastic Net, are sparse models, in which the number of influential predictors (with non-zero coefficients) is kept small. As a consequence, these ML methods tend to omit statistically significant features when they are correlated to each other in favor of a smaller subset of features with optimal predictive power. Nonetheless, in our meta-analysis of CD, the feature weights derived from multivariable modeling corresponded well to the univariate associations, and also showed some consistency across the four studies in which clear CD associations could be detected and an accurate ML model trained (Fig. 6a, right panel). Taken together, these results demonstrate that SIAMCAT could be a tool of broad utility for consolidating microbiome-disease associations and biomarker discovery by leveraging a large amount of metagenomic data becoming available for ML-based analyses.

## Discussion

The rising interest in clinical microbiome studies and microbiome-derived diagnostic, prognostic, and therapeutic biomarkers also calls for more robust analysis procedures. An important step in that direction is the development of freely available, comprehensive, and extensively validated analysis workflows that make complex ML procedures

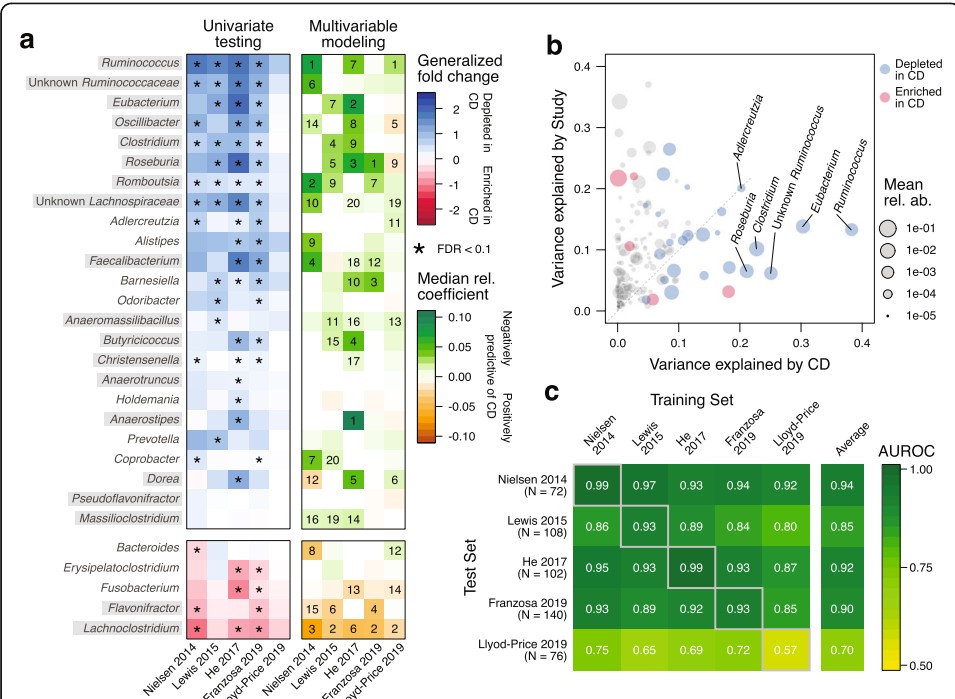

**Fig. 6** Meta-analysis of CD studies based on fecal shotgun metagenomic data. **a** Genus-level univariate and multivariable associations with CD across the five included metagenomic studies. The heatmap on the left side shows the generalized fold change for genera with a single-feature AUROC higher than 0.75 or smaller than 0.25 in at least one of the studies. Associations with a false discovery rate (FDR) below 0.1 are highlighted by a star. Statistical significance was tested using a Wilcoxon test and corrected for multiple testing using the Benjamini-Hochberg procedure. Genera are ordered according to the mean fold change across studies, and genera belonging to the Clostridiales order are highlighted by gray boxes. The right side displays the median model weights for the same genera derived from Elastic Net models trained on the five different studies. For each dataset, the top 20 features (regarding their absolute weight) are indicated by their rank. **b** Variance explained by disease status (CD vs controls) is plotted against the variance explained by differences between studies for individual genera. The dot size is proportional to the mean abundance, and genera included in **a** are highlighted in red or blue. **c** Classification accuracy as measured by AUROC is shown as a heatmap for Elastic Net models trained on genus-level abundances to distinguish controls from CD cases. The diagonal displays values resulting from cross-validation (when the test and training set are the same), and off-diagonal boxes show the results from the study-to-study transfer of models

available to non-experts, ideally while safeguarding against statistical analysis pitfalls. Designed with these objectives in mind, SIAMCAT provides a modular analysis framework that builds on the existing R-based microbiome analysis environment: data integration from DADA2 [98] or phyloseq [69] is straightforward since SIAMCAT internally uses the phyloseq object. ML algorithms and procedures in SIAMCAT interface to the mlr package [70], a general-purpose ML library. Since the multitude of ML algorithms, workflow options, and design choices within such a general package can make ML workflow design challenging for non-experts, SIAMCAT mainly aims to enable users to apply robust and validated ML workflows to their data with preprocessing and normalization options tailored to the characteristics of microbiome data. At the same time, SIAMCAT allows advanced users to flexibly set up and customize more complex ML procedures, including non-standard cross-validation splits for dependent measurements and supervised feature selection methods that are properly nested into cross-validation (Fig. 3). Further developments of the package are planned to

accommodate the rapidly changing needs of the microbiome research community, and updates will be published in accordance with the established Bioconductor release schedule.

To showcase the power of ML workflows implemented in SIAMCAT, we performed a meta-analysis of human gut metagenomic studies at a considerably larger scale than previous efforts [17–22] (see Fig. 4). It importantly encompassed a large number of diseases as well as different taxonomic and functional profiles as input that were derived from different metagenomic sequencing techniques (16S rRNA gene and shotgun metagenomics sequencing) and profiling tools. Consequently, these benchmarks are expected to yield much more robust and general results than those from previous studies [17–22]. In our exploration of more than 7000 different parameter combinations per classification task (see the "Methods" section), we found the Elastic Net logistic regression algorithm to yield the highest cross-validation accuracies on average, albeit requiring the input data to be appropriately normalized (see Fig. 4 and Additional file 1: Figure S8). Compared with the choice of normalization method and classification algorithm, other parameters had a considerably lower influence on the resulting classification accuracy. SIAMCAT's functionality to robustly fit statistical microbiome models and evaluate their performance will enable comparison to established diagnostic biomarkers [8] as an important prerequisite for further translation of microbiome research into the clinic.

To help resolve the debate about spurious associations and reproducibility issues in microbiome research [99], meta-analyses are crucial for the validation of microbiome biomarkers [37, 38]. However, we found that ML models have substantial problems with type I error control (> 2-fold increase in FPR) and disease specificity (> 2.5-fold elevated FPR) when naively transferred across studies. We propose measures to detect these issues, which, if more widely adopted, could help to more precisely characterize them and their underlying causes. To address them, we introduce the control augmentation strategy, which greatly improved the cross-study portability of ML models. Being the first attempt to overcome study heterogeneity for improved cross-study model application, our work will hopefully stimulate further developments, which could easily be evaluated on the provided datasets. However, all such ML meta-analyses are limited by biological and clinical differences between studies [91], which will have to be addressed by better reporting standards [100]. Within these limitations, our ML meta-analysis datasets could become a valuable community resource for method development, systematic assessment of disease similarities, and further exploration of globally applicable microbiome biomarkers including validation of their disease specificity.

Using model introspection after control augmentation, we could revisit the question if microbiome alterations are specific to a given disease or more general hallmarks of dysbiosis [20]. In general, we found depletion of beneficial bacteria to be more often shared across several diseases (e.g., *Anaerostipes* or *Romboutisa*), in particular, in the subtypes of IBD. Conversely, disease-enriched bacteria were more often specific to a given disease. This could mean that some disease-specific microbiome alterations may reflect pathogens or pathobionts acting either as etiological agents or exploiting specific disease-related changes in the intestinal milieu. As examples of disease-specific markers, *Parvimonas* spp. are predictive for colorectal cancer, which is consistent with

mechanistic work demonstrating this species to accelerate proliferation and cancer development both in vitro and in vivo [101]. Similarly, a putative link between oral *Veillonella* spp. and liver cirrhosis severity has been reported in the context of proton-pump inhibitor therapy [102], potentially enabled by increased transmission from the oral to the gut microbiome [78]. Other taxa showing a broader disease spectrum, such as *Fusobacterium* spp., have been extensively studied both in the context of CRC [103] and in IBD [104] using cellular and animal models. However, firmly establishing disease specificity or disease spectra for microbial biomarkers will be difficult to achieve in preclinical studies but require large patient cohorts. Nonetheless, our analyses generated candidates of both shared and disease-specific gut microbial biomarkers to guide further investigations of specific hypotheses on their ecological roles.

Although the analyses presented here are focused on human gut metagenomic datasets with disease prediction tasks, SIAMCAT is not restricted to these. It can also be applied to other tasks of interest in microbiome research, e.g., for investigating the effects of medication (see Additional file 1: Figure S2). Metagenomic or metatranscriptomic data from environmental samples can also be analyzed using SIAMCAT, e.g., to understand the associations between community composition and transcriptional activity of the ocean microbiome with physicochemical environmental properties (see Additional file 1: Figure S16 for an example [105]) highlighting that SIAMCAT could be of broad utility in microbiome research.

## Methods

### Implementation

SIAMCAT is implemented as an R package with a modular architecture, allowing for a flexible combination of different functions to build ML and statistical analysis workflows (see the "Code box" section). The output of the functions (for example, the feature matrix after normalization) is stored in the SIAMCAT object, which is an extension of the *phyloseq* object that contains the raw feature abundances, meta-variables about the samples, and other optional information (for example, a taxonomy table or a phylogenetic tree) [69]. The label defining the sample groups for comparison is then derived from a user-specified meta-variable or an additional vector. ML models are trained using the *mlr* infrastructure as an interface to the implementations of different ML algorithms in other R packages [70]. SIAMCAT is available under the GNU General Public License, version 3.

### Code box

Given two R objects called `feat` (relative abundance matrix) and `meta` (meta-variables about samples as a dataframe, containing a column called `disease` which encodes the label), the entire analysis can be conducted with a few commands (more extensive documentation can be found online in the SIAMCAT vignettes).

```
sc.obj <- siamcat(feat=feat, meta=meta, label='disease')
sc.obj <- filter.features(sc.obj, filter.method = 'abundance')
sc.obj <- check.associations(sc.obj,
    fn.plot = 'associations_plot.pdf') # produces Fig. 1b
check.confounders(sc.obj,
    fn.plot = 'confounder_plot.pdf') # produces Fig. 1c
sc.obj <- normalize.features(sc.obj, norm.method = 'log.std')
sc.obj <- create.data.split(sc.obj)
sc.obj <- train.model(sc.obj, method='lasso')
sc.obj <- make.predictions(sc.obj)
sc.obj <- evaluate.predictions(sc.obj)
model.evaluation.plot(sc.obj,
    fn.plot = 'evaluation.pdf') # produces Fig. 1d
model.interpretation.plot(sc.obj, consens.thres = 0.8,
    fn.plot = 'interpretation.pdf') # produces Fig. 1e
```

### Included datasets and microbiome profiling

In this study, we analyzed taxonomic and functional profiles derived with different profiling tools from several metagenomic datasets (see Additional file 2: Table S1). Taxonomic profiles generated using the RDP classifier [83] on the basis of 16S rRNA gene sequencing data were downloaded from a recent meta-analysis by Duvallet et al. [20] and summarized at the genus level. MetaPhlAn2 [84] and HUMAnN2 [86] taxonomic and functional profiles were obtained from the *curatedMetagenomicsData* R package [72] for all human gut datasets within the package that contained at least 20 cases and 20 controls. MetaPhlAn2 profiles were filtered to contain only species-level microbial taxa.

Additional datasets were profiled in-house with the following pipeline implemented in *NGless* [106]: after preprocessing with MOCAT2 [107] and filtering for human reads, taxonomic profiles were generated using the mOTUsv2 profiler [85], and functional profiles were calculated by first mapping reads against the integrated gene catalog [108] and then aggregating the results by eggNOG orthologous groups [87].

Additionally, genus-level taxonomic profiles from the TARA Oceans microbiome project [105] were used for two different classification tasks: to classify samples from polar and non-polar ocean regions and to classify samples based on their iron concentration at a depth of 5 m (high vs low iron content).

### Primary package outputs and confounder analysis

To illustrate the main outputs of SIAMCAT, we analyzed the taxonomic profiles from a metagenomic study of IBD [27] included in the *curatedMetagenomicsData* R package

[72]. For the analyses presented in Fig. 1, we restricted the dataset to control samples from Spain and cases with UC, since the two IBD subtypes included in the dataset (ulcerative colitis and Crohn's disease) are very different from one another in terms of the associated changes in the gut microbiome composition (see the SIAMCAT vignettes for more information or the "Code box" section for an outline of the basic SIAMCAT workflow.

To demonstrate how SIAMCAT can aid in confounder detection, we used the same dataset but this time included the Danish control samples in order to explore potential confounding by differences between samples collected and processed in these two countries. The analyses presented in Fig. 2 have all been conducted with the respective functions of SIAMCAT (see SIAMCAT vignettes).

### Machine learning hyperparameter exploration

To explore suitable hyperparameter combinations for ML workflows, we trained an ML model for each classification task and each hyperparameter combination. By hyperparameter, we mean configuration parameters of the workflow, such as normalization parameters, tuning parameters controlling regularization strength, or properties of the external feature selection procedure in contrast to model parameters fitted during the actual training of the ML algorithms. Specifically, we varied the filtering method (no data filtering; prevalence filtering with 1%, 5%, 10% cutoffs; abundance filtering with 0.001, 0.0001, and 0.0001 as cutoffs; and a combination of abundance and prevalence filtering), the normalization method (no normalization beyond the total sum scaling, log-transformation with standardization, rank-transformation with standardization, and centered log-ratio transformation), the ML algorithm (LASSO, Elastic Net, and random forest classifiers), and feature selection regimes (no feature selection and feature selection based on generalized fold change or based on single-feature AUROC; cutoffs were 25, 50, 100, 200, and 400 features for taxonomic profiles and 100, 500, 1000, and 2000 features for functional profiles). To cover the full hyperparameter space, we therefore trained 7488 models for taxonomic and 3168 models for functional datasets for each classification task.

To determine the optimal AUROC across input types (shown in Fig. 4), we calculated for each individual parameter combination the mean AUROC across all classification tasks with a specific type of input. Different feature filtering procedures could lead to cases in which the feature selection cutoffs were larger than the number of available features after filtering, therefore terminating the ML procedure. For that reason, we only considered those parameter combinations that did produce a result for all classification tasks with the specific type of input data.

To compare the importance of feature filtering, feature selection, normalization method, and ML algorithm on classification accuracy, we trained one linear model per classification task predicting the AUROC values from those variables. We then partitioned the variance attributable to each of these variables by calculating type III sums of squares using the Anova function from the car package in R [109].

In order to contrast the class separation of samples in distance space with the classification performance achieved by ML algorithms (see Additional file 1: Figure S6), we designed a distance-based measure of separation. For each dataset, we determined the distances between all pairs of samples within a class as well as all pairs of samples

between classes and then calculated an AUROC value based on these two distributions. This distance-based measure effectively quantifies to what extent samples are closest to other samples from the same class (i.e., cluster together) and hence corresponds well to the visual separation of classes in ordination space (see Additional file 1: Figure S6).

### Model transfer, cross-study portability, and prediction rate for other diseases

To assess cross-study portability and prediction rate for other diseases, ML models were applied to external datasets using the *make.predictions* function in SIAMCAT. In short, the function uses the normalization parameters of the discovery dataset to normalize the external data in a comparable way and then makes predictions by averaging the results of the application of all models of the repeated cross-validation folds to the normalized external data.

Cross-study portability is then calculated by comparing the predictions for cases in the discovery datasets and controls in the external dataset. First, the AUROC between these two prediction vectors is calculated, and values below 0.5 (when the predictions on controls in the external dataset are higher than predictions on cases in the discovery dataset) are set to 0.5. Cross-study portability is then defined as $(|0.5 - AUROC|)*2$ so that it afterwards ranges from 0 (no separation between cases and external controls or higher predictions on external controls) to 1 (perfect separation between cases and external controls).

To calculate the prediction rate for other diseases (or the same disease) on external datasets, a cutoff on the (real-valued) predictions is chosen so that the FPR in the discovery dataset is 0.1. Based on this cutoff, the external predictions are evaluated as positive (diseased) or negative predictions, and a detection rate corresponding to the fraction of positive predictions is determined.

### Training Elastic Net models with control augmentation

To train models with the control augmentation strategy, we used the data from cohort microbiome studies as additional control samples [93–95]. Repeated measurements for the same individual were removed in the case of Zeevi et al. [93]. For each training set in the repeated cross-validation, we increased the number of control samples 5-fold by randomly sampling the appropriate number of controls (in a balanced manner between datasets to avoid overrepresentation of the larger external cohorts). Before addition, the additional control samples were normalized using the normalization parameters of the discovery set. Due to the introduction of additional variability, the control-augmented Elastic Net models were trained with a pre-set alpha value of 0.5 to ensure the stability of the model size.

To compare the predictors across different diseases, model weights of the control-augmented models were transformed into relative weights by dividing by the sum of absolute coefficient values. Then, models from the same disease were averaged. Predictors (that is, mOTUs) were selected for display in Fig. 5f, if they (i) cumulatively contributed more than 50% of the mean relative disease model, (ii) their individual weights were bigger than 1%, and (iii) the genus annotation had an unambiguous NCBI taxonomy.

### Illustration of common pitfalls in machine learning procedures

To demonstrate how naive sequential application of supervised feature selection and cross-validation might bias performance estimations, we trained LASSO ML models to distinguish colorectal cancer cases from controls based on MetaPhlAn2-derived species abundance profiles using the dataset with the handle *ThomasAM_2018a* [38] obtained through the *curatedMetagenomicsData* R package [72]. For the incorrect procedure of feature selection, single-feature AUROC values were calculated using the complete dataset (inverted for negatively associated features). Then, the features with the highest AUROC values were selected for model training (number depending on the cutoff). In contrast, the correct procedure implemented in SIAMCAT excludes the data in the test fold when calculating single-feature AUROC values; instead, AUROC values are calculated on the training fold only. To test generalization to external data, the models were then applied to another colorectal cancer metagenomic study [8] available through the *curatedMetagenomicsData* R package (also see the SIAMCAT vignette: holdout testing).

To illustrate the problem arising when combining naive cross-validation with dependent data, we used the Crohn's disease (CD) datasets used in the meta-analysis described below. We first subsampled the iHMP dataset [30] to five repeated measurements per subject, as some subjects had been sampled only five times and others more than 20 times. Then, we trained LASSO models using both a naive cross-validation and a cross-validation procedure in which samples from the same individual were always kept together in the same fold. External generalization was tested on the other four CD datasets described below.

### Meta-analysis of Crohn's disease metagenomic studies

For the meta-analysis of Crohn's disease gut microbiome studies, we included five metagenomic datasets [5, 27–30] that had been profiled with the mOTUs2 profiler [85] on the genus level. While some datasets contained both UC and CD patients [5, 27, 30], other datasets contained only CD cases [28, 29]. Therefore, we restricted all datasets to a comparison between only CD cases and control samples, since the two subtypes of IBD are very different from each other.

For training of ML models, we blocked repeated measurements for the same individual when applicable [27, 28, 30]; specifically for the iHMP dataset [30], we also subsampled the dataset to five repeated measurements per individual to avoid biases associated with differences in the number of samples per individual. For external validation testing, we completely removed repeated measurements in order not to bias the estimation of classification accuracy.

To compute association metrics and to train and evaluate ML models, each dataset was encapsulated in an individual SIAMCAT object. To produce the plot showing the variance explained by label vs the variance explained by study, all data were combined into a single SIAMCAT object. The code to reproduce the analysis can be found in the SIAMCAT vignettes.

### Supplementary Information

**Additional file 1: Figure S1.** SIAMCAT reproduces the results of previous machine learning meta-analyses. **Figure S2.** SIAMCAT can detect confounding factors such as metformin treatment. **Figure S3.** Metagenomic samples are more similar within subjects than across subjects. **Figure S4.** Large-scale application of the SIAMCAT machine learning workflow to human gut metagenomic disease association studies in the curatedMetagenomicData package. **Figure S5**. Dataset size relates to classification accuracy and the AUROC confidence interval. **Figure S6**. Machine learning can distinguish group differences even when samples can not be separated based on common ecological distances. **Figure S7**. Classification accuracy is not impacted by choice of profiler. **Figure S8**. Influence of feature selection cutoff and normalization method on classification accuracy. **Figure S9**. Baseline evaluation of cross-study transfer of machine learning models via AUROC and false-positive rate. **Figure S10.** Measures for extended evaluation of cross-study application of machine learning models. **Figure S11**. Naive machine learning models show lower cross-study portability when applied to external datasets compared to control-augmented models. **Figure S12**. Naive machine learning models make a high level of false predictions on external datasets. **Figure S13**. Control-augmentation strategy generally improves model transfer without a strong dependence on the type and number of control samples. **Figure S14**. Datasets cluster by disease, both when considering machine learning model weights or associations. **Figure S15.** Machine learning model weights reveal shared and disease-specific predictors. **Figure S16**. SIAMCAT can be applied to metagenomic and metatranscriptomic measurements from environmental samples.

**Additional file 2: Supplementary Tables. Table S1**. Information about included datasets.

**Additional file 3.** Review history.

## Acknowledgements

We are grateful to Mike Smith, Paul I. Costea, and Kersten Breuer for the helpful discussions and advice on the implementation of SIAMCAT. We thank members of the Zeller, Sunagawa, and Bork group for the fruitful discussions and the EMBL Information Technology Core Facility for the support with high-performance computing.

## Peer review information

## Review history

The review history is available as Additional file 3.

## Authors' contributions

G.Z. conceived the study and prototyped the software. GZ., SS., and P.B. supervised the work. K.Z., J.W., and G.Z. implemented the software package with contributions from M.E., N. K, and E.K. J.W. and G.S. acquired the metagenomic data and/or performed the taxonomic and functional profiling. J.W., G.Z., and N.K. designed and performed the statistical analyses. J.W. and G.Z. designed the figures with help from N.K., M.E., and E.K. J.W., G.Z., and S.S. wrote the manuscript with contributions from P.B., M.E., N.K., G.S., E.K., and K.Z. All authors discussed and approved the final manuscript.

## Funding

We acknowledge funding from EMBL, ETH (PHRT no. 521 to S.S.), the Federal Ministry of Education and Research (BMBF; the de. NBI network no. 031A537B to P.B. and grant no. 031L0181A to G.Z. and P.B.), the Deutsche Forschungsgemeinschaft (DFG, German Research Foundation no. 395357507 – SFB 1371 to G.Z.), and the Helmut Horten Foundation (to S.S.). Open Access funding enabled and organized by Projekt DEAL.

## Availability of data and materials

Raw metagenomics data are available from ENA (see Additional file 2: Table S1 for the identifiers of included datasets). All taxonomic and functional profiles used as input for the presented analyses are available in a Zenodo repository (see either https://doi.org/10.5281/zenodo.4454489 [110]), and the code to reproduce the analysis can be found in the dedicated GitHub repository (https://github.com/zellerlab/siamcat_paper [111]).
The code for SIAMCAT can be found on GitHub (https://github.com/zellerlab/siamcat [112]) and on Bioconductor under https://doi.org/10.18129/B9.bioc.SIAMCAT.
The source code for SIAMCAT and the code to reproduce the analysis presented in this paper are both available from Zenodo (see https://doi.org/10.5281/zenodo.4457522 [113]) under the GPL-3 license.

# Declarations

## Ethics approval and consent to participate

Not applicable.

## Competing interests

The authors declared that they have no competing interests.

## Author details

[1]Structural and Computational Biology Unit, European Molecular Biology Laboratory (EMBL), 69117 Heidelberg, Germany. [2]Present Address: Clinical Microbiomics A/S, Ole Maaløes Vej 3, 2200 København, Denmark. [3]Present Address: Experimental and Clinical Research Center (ECRC) of the Max Delbrück Center for Molecular Medicine and Charité University Hospital, 13125 Berlin, Germany. [4]Department CIBIO, University of Trento, 38123 Trento, Italy. [5]Department of

Biology, Institute of Microbiology and Swiss Institute of Bioinformatics, ETH Zürich, 8093 Zürich, Switzerland. ⁶Molecular Medicine Partnership Unit, Heidelberg, Germany. ⁷Max Delbrück Centre for Molecular Medicine, 13125 Berlin, Germany . ⁸Department of Bioinformatics, Biocenter, University of Würzburg, 97074 Würzburg, Germany.

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

## 
