## [**Additional file 3.** Review history. · Genome Biology]

Review History

First round of review

Reviewer 1

Were you able to assess all statistics in the manuscript, including the appropriateness of statistical tests used? Yes, and I have assessed the statistics in my report.

Were you able to directly test the software? No.

Comments to author:

This new R package by Zeller and colleagues aim to provide a general and easy framework for carrying out machine learning with microbiome data, and this manuscript describes relatively thoroughly the rationale, design and how-to regarding the package. In my personal opinion, the logic and methods chosen are appropriate and the package will surely be very instrumental for the larger scale studies.

The description on the methodology is quite clear, but I do suggest that the authors work on the writing a little bit. For instance, at the beginning of Results section as Figure 1 (d/e) and Figure 2 already contains real data, it would be better to have some of background information regarding those used data in the main text (instead of figure legends) for better comprehension. Also, move table 1 to the front, so readers can infer the studies most easily.

I would also suggest the authors discuss a little bit more regarding the technicality vs biological meaningfulness, which I feel is not adequate in the current Discussion section, a method can be very logical, but it has to solve some real issues. Here the authors discovered some signatures regarding CRC, CD and other diseases, but it is not thoroughly compared to published studies/mechanistic investigations of whether those bacteria ARE indeed the major biomarkers, for instance Parvimonas in CRC, is it supported by many studies?

Reviewer 2

Were you able to assess all statistics in the manuscript, including the appropriateness of statistical tests used? There are no statistics in the manuscript.

Were you able to directly test the software? Yes.

Comments to author:

In this manuscript, Wirbel et al. present the SIAMCAT software package that facilitates ML interrogate of microbiome data in R, and use it to demonstrate several pitfalls of ML analysis on microbiome data, and to investigate the dependence of ML analysis of microbiome data on various parameters and differences in studies. This manuscript and software package directly addresses important and highly relevant questions in current microbiome science. I will briefly survey the contributions of this manuscript, and while I will provide critiques of each (as in the nature of a review) I think that overall this software and analysis is a timely and worthwhile contribution to the literature that has the potential to advance future microbiome science as well.

Contribution 1: The SIAMCAT R software package. SIAMCAT integrates with microbiome-specific R data manipulation (phyloseq). The authors state that SIAMCAT is unique in providing easy-to-use and validated ML for microbiome analysis. However, the defined input to SIAMCAT is very generic (feature-

table plus case definition and covariates) and thus the real comparison seems to be to much more generic ML software. Where does SIAMCAT fit in feature-wise compared to relevant ML packages? (e.g. caret, but many others). Also, if software is the main contribution here, the format isn't ideal (code is separated from the demonstrations). The implementation and documentation is good, e.g. in Bioconductor for >2 years. But... what is the long-term community interaction plan? In particular user support, and development including community input? Given that the primary(?) contribution described in this paper is the software, these questions are critical.

Contribution 2: Evaluation of cross-study predictivity of ML methods. This is a major issue that has not yet received enough attention in the microbiome area, and I applaud the authors for taking it on. Technical differences are known to be major issues, and what use is a ML method that doesn't translate to new data? A major shortcoming, however, is that this paper largely ignores major non-microbiome-specific issues that can reduce cross-study concordance such as differences in study population and the definition of the clinical phenotype being studied. This is even shown in the nationality example, but then ignored thereafter! However, the most important problem with the current study is the way in which performance between studies is defined. I was not convinced of the validity/relevance of "cross-study portability" as defined in this paper.. Why do I care how accurate my ML method to diagnose UC is on other diseases? If it's accurate for UC, who cares if it also has some level of prediction for e.g. T2D? Why shouldn't we be evaluating something obvious, like prediction accuracy of a model for disease XX trained in study YY on study ZZ (that also dealt with disease XX)?

Contribution 3: The authors present a "control augmentation strategy" for improving ML models predicting disease states from microbiome data. However, there are major shortcomings in the data presented if this is to be interpreted as a serious contribution from this paper. The authors chose 3 studies to include augmented controls from, but why those studies were chosen is unclear. Does the choice of augmenting studies matter? It seems like it should... a "control" in one study is not the same as a control in another. This brings up an important question: Why should a control in a T2D study be counted as a control in a e.g. Crohn's study? Or... if it shouldn't, where are the lines? This method is described, but not evaluated in a way that makes it seem useful at this point beyond the paper.

Contribution 4: Discussion and demonstration of some common pitfalls in a microbiome context. As authors note, these are widely known and discussed in stats/ML literature, but there is value in showing them in a subject-specific context as well. I commend the authors on making the effort to both demonstrate these issues with microbiome data, and to make the demonstrations accessible for others to replicate on their own.

More minor itemized comments:

P5L27-30: It's important to also discuss metagenomics bias here. See for example recent Willis lab papers (e.g. <https://doi.org/10.1101/761486> and <https://doi.org/10.7554/eLife.46923.001>).

P5L43-44: I agree that (as far as I know) there is no microbiome-specific ML package, and that given the increased usage of such methods in microbiome research that could be very useful.

P5L1-3: Agree completely. This is a huge problem.

P6L26-30: These are very generic inputs, that seem to place SIAMCAT in the realm of ML methods generally. There doesn't seem to be any utilization of microbiome-specific features such as the hierarchical taxonomy or phylogeny data types.

P6L41-43: This sort of functionality seems available in a number of other places (e.g. metagenomeSeq, or

DESeq2).

P6L46: What does SIAMCAT add relative to other dedicated ML packagers (caret is one, but there are many others)?

P9L7-9: This is a very nice proof-of-principle on this issues. Question: Can SIAMCAT models include confounders like this?

P10L34-35: This is useful and has value as a subject-specific demonstration of these issues. But this isn't really about SIAMCAT the software package.

P11L58-59: This is an example of where it's unfortunate the code is not more well integrated with the text.

P12L57: Should say right away (given previous text) that the rest of this section is all discussing intra-study results.

P12L59: Is AUROC >0.75 equivalent to "accurately distinguish"? Not really.

P13L5-6;L14: "accurate microbiome signatures" are not shown here.

P13L17-18: "offers implementations..." Is SIAMCAT implementing ML algorithms? I don't think so, and this is the point to make that clear and also mention the implementations the package is relying on.

P13L32-33: We see the same with the RF classifier being good in the sense of being robust to microbiome normalization!

P13L42-43: Language is a bit strong for me. E.g. "Comprehensive" is too much IMO given the evaluations conducted.

P14L46-47: Probably an overestimate, no, given the systematic issues introduced above?

P15L11-12: This is a very interesting question.

P15L15-16: huh?

P15L15-16: Why? Why isn't cross-study predictivity being evaluated? Instead of this... weird choice.

P15L27-28: What about other factors, like population heterogeneity (as was so ably demonstrated above) or the specifics of how cases and controls were defined? There is a huge background in clinical meta-analyses that demonstrates the importance of these sorts of issues that seems to just be ignored here. This may be microbiome focused, but the microbiome is not the extent of the issues one must confront in doing clinically relevant microbiome science!

P15L44-45: Have to define cross-study "portability" before this. See also previous concerns about this concept.

Fig5c: Is "prediction rate for other diseases" defined yet?

P17L46-48: Is this still kind of bad, compared to other biomarkers in clinical usage?

P20L4-5: Not sure anything here is improving "standardization".

P20L20-21: If this is a software paper, this information needs to be way earlier. To be honest, this was a hard to shake discomfort with this paper... so much of what is was doing was useful and needed in the microbiome field, but it didn't seem to completely acknowledge what is was built on (e.g. mlr, but more than that).

P20:L59: "disease-specificity". See also above, but... by who I care? If I predict T2D with 100% accuracy, why do I care how that same model does in predicting obesity?

Authors Response

Point-by-point responses to the reviewers' comments:

Reviewer #1: This new R package by Zeller and colleagues aim to provide a general and easy framework for carrying out machine learning with microbiome data, and this manuscript describes relatively thoroughly the rationale, design and how-to regarding the package. In my personal opinion, the logic and methods chosen are appropriate and the package will surely be very instrumental for the larger scale studies.

We thank the reviewer for their positive assessment of our manuscript.

The description on the methodology is quite clear, but I do suggest that the authors work on the writing a little bit. For instance, at the beginning of Results section as Figure 1 (d/e) and Figure 2 already contains real data, it would be better to have some of background information regarding those used data in the main text (instead of figure legends) for better comprehension. Also, move table 1 to the front, so readers can infer the studies most easily.

We apologize for a lack of detail in the main text regarding the real-world demonstrative example dataset used in Figure 1 and 2.

Actions taken: We re-phrased the paragraph that introduces the example dataset to provide more biological context, see P5L15-17. We also introduced Table 1 earlier in the main text, see P4.

I would also suggest the authors discuss a little bit more regarding the technicality vs biological meaningfulness, which I feel is not adequate in the current Discussion section, a method can be very logical, but it has to solve some real issues. Here the authors discovered some signatures regarding CRC, CD and other diseases, but it is not thoroughly compared to published studies/mechanistic investigations of whether those bacteria ARE indeed the major biomarkers, for instance Parvimonas in CRC, is it supported by many studies?

We agree with the reviewer that biological and technical aspects should have been better balanced in the Discussion and that mechanistic evidence for etiological contributions of these microbial biomarkers, in the cases in which it exists, warrants discussion. In the revised manuscript we now mention a few examples (including Parvimonas in CRC), but would also like to note that this cannot be done comprehensively given the many conditions and data sets included in our metaanalysis. Moreover, as our emphasis is on differentiating between disease-specific versus broader markers of dysbiosis, it is quite difficult to compare our results to mechanistic work, as is now also acknowledged in the Discussion.

Actions taken: We included an additional paragraph in the Discussion section to contextualise our results with mechanistic studies of putative disease-specific biomarkers uncovered in our ML meta-analysis, see P20L10-39.

Reviewer #2: In this manuscript, Wirbel et al. present the SIAMCAT software package that facilitates ML interrogation of microbiome data in R, and use it to demonstrate several pitfalls of ML analysis on microbiome data, and to investigate the dependence of ML analysis of microbiome data on various parameters and differences in studies. This manuscript and software package directly addresses important and highly relevant questions in current microbiome science. I will briefly survey the contributions of this manuscript, and while I will provide critiques of each (as in the nature of a review) I think that overall this software and analysis is a timely and worthwhile contribution to the literature that has the potential to advance future microbiome science as well.

We thank the reviewer for their positive comments on our work.

Contribution 1: The SIAMCAT R software package. SIAMCAT integrates with microbiome-specific R data manipulation (phyloseq). The authors state that SIAMCAT is unique in providing easy-to-use and validated ML for microbiome analysis. However, the defined input to SIAMCAT is very generic (feature-table plus case definition and covariates) and thus the real comparison seems to be to much more generic ML software. Where does SIAMCAT fit in feature-wise compared to relevant ML packages? (e.g. caret, but many others). Also, if software is the main contribution here, the format isn't ideal (code is separated from the demonstrations). The implementation and documentation is good, e.g. in Bioconductor for >2 years. But... what is the longterm community interaction plan? In particular user support, and development including community input? Given that the primary(?) contribution described in this paper is the software, these questions are critical.

We agree with the reviewer that the paper would be improved if the code could be better integrated into the main text and we thank them for this suggestion. In contrast to general-purpose ML libraries, such as caret or mlr (which is internally used by SIAMCAT), SIAMCAT mainly aims to enable users without a lot of prior experience with ML to apply robust ML pipelines to their data, while offering microbiome-specific choices for data filtering and normalization. We feel that a major contribution of our work is that we extensively validated the many parameter choices in these workflows on a broad range of microbiome studies.

We also agree with the reviewer that the disconnect between code and demonstrations can become problematic as the software evolves further. We have therefore greatly expanded the vignettes available with the SIAMCAT Bioconductor package and will update these together with the code to keep these analyses "alive" for future reproduction and customisation. We included a new vignette for each of the analyses shown in the figures 1, 2, 3, and 6.

Lastly, we thank the reviewer for their suggestion to elaborate on the community interaction plan and possible future developments. Further extensions are planned or already ongoing as grant-funded research projects within our group. Additionally, we welcome input and contributions from the community through the SIAMCAT Github repository (<https://github.com/zellerlab/siamcat>), which also offers the possibility of bug reports and feature requests via the Issues page. Additional user support is provided via Stackoverflow, and a mailing list. We will update the SIAMCAT package in accordance with the Bioconductor release schedule (see P19L19-21).

Actions taken: We extended the first paragraph of the discussion to consider potential community input and further developments to the package. Additionally, we included a sentence regarding a conceptual comparison to caret or bare mlr ML pipelines (P19L11-16).

Contribution 2: Evaluation of cross-study predictivity of ML methods. This is a major issue that has not yet received enough attention in the microbiome area, and I applaud the authors for taking it on. Technical differences are known to be major issues, and what use is a ML method that doesn't translate to new data? A major shortcoming, however, is that this paper largely ignores major non-microbiome-specific issues that can reduce cross-study concordance such as differences in study population and the definition of the clinical phenotype being studied. This is even shown in the nationality example, but then ignored thereafter! However, the most important problem with the current study is the way in which performance between studies is defined. I was not convinced of the validity/relevance of "cross-study portability" as defined in this paper.. Why do I care how accurate my ML method to diagnose UC is on other diseases? If it's accurate for UC, who cares if it also has some level of prediction for e.g. T2D? Why shouldn't we be evaluating something obvious, like prediction accuracy of a model for disease XX trained in study YY on study ZZ (that also dealt with disease XX)?

We feel encouraged by the reviewer recognising the relevance of our analyses in general. We also agree with the reviewer that the discussion of these concepts needed to be further developed in our manuscript. In particular,

1. we first acknowledge that our definition of study heterogeneity may have been too narrow and we therefore extended it as suggested in the revised manuscript (P13). We also discuss the limitations arising from some of these sources of study heterogeneity for any meta-analysis (P20).

2. In the revised manuscript we also much more carefully motivate and explain the relevance of cross-study portability of ML models and the measures we chose to assess it. Our measure of cross-study portability is essentially a ROC analysis (rescaled AUROC) in which external controls are used instead of those from the crossvalidation data set. It simply assesses the separation between cases in the original dataset and control samples in the external test set (see newly added SFig9). This choice enables its broad application across datasets from many (different) conditions. Interestingly, we observed cases in which the predictions for the control samples from the test set were even higher than the predictions for the cases in the original dataset (see SFig9, Panel d), illustrating how little separation between control samples and cases can be observed during naive model transfer in extreme cases. Our measure of disease specificity is a straightforward calculation of the false-positive rate on samples from other conditions.

3. Finally, the reviewer's questions about disease specificity prompted us to better motivate why we feel that this is relevant, in particular for potential future application in population screening for conditions that could differ substantially in their prevalence. As explained in the revised text (P13), we believe that one should care if an (accurate) CRC model shows elevated false positive rates for samples from other diseases. This would indicate that the microbiome changes captured by the model are not specific to this disease. Consider the case when a microbiome-based test for a rare disease (for example, CRC) has a high false positive rate in (very common) T2D cases. Using such a test as a population screening tool would create a vast number of false positives resulting in very low precision.

Actions taken: -We mention additional sources of study heterogeneity highlighted by the reviewer (P13L9-12) -We included an additional supplementary figure (SFig9) to better explain and motivate

the measures we calculate when transferring models across datasets and extended on the paragraph introducing and explaining these measures (see P13L19-38). -Additionally, we also included a discussion of the limitations some of these study differences pose on any attempt of cross-study ML analyses in the main text, (P20L1- 6).

Contribution 3: The authors present a "control augmentation strategy" for improving ML models predicting disease states from microbiome data. However, there are major shortcomings in the data presented if this is to be interpreted as a serious contribution from this paper. The authors chose 3 studies to include augmented controls from, but why those studies were chosen is unclear. Does the choice of augmenting studies matter? It seems like it should... a "control" in one study is not the same as a control in another. This brings up an important question: Why should a control in a T2D study be counted as a control in a e.g. Crohn's study? Or... if it shouldn't, where are the lines? This method is described, but not evaluated in a way that makes it seem useful at this point beyond the paper.

We acknowledge the shortcomings of our presented results and thank the reviewer for their criticism to improve our manuscript.

To answer the question, why we chose those three studies for control-augmentation, we selected cohort-studies (including mostly asymptomatic subjects and diseased subjects only at the population-level prevalence) in order to have a large sample size (>250 samples, needed to augment every study in our meta-analysis) and to minimize label noise due to other conditions. We are fully aware that "control" is not a clear concept and its definition varies greatly across studies. Nonetheless we are convinced that it is useful at an operational level to enrich for asymptomatic individuals, therefore reducing bias that could result from unintended comparisons to patients with a different disease.

To investigate the sensitivity of the control augmentation strategy to choice of controls and parameters, we augmented our datasets with a different ratio of control samples (from the same cohort studies) or with control samples from a different set of studies (either a totally different set of controls or controls chosen randomly from the datasets in the meta-analysis). This led to very similar results in terms of cross-study portability and disease specificity across the same set of evaluations (see newly added SFig12). Here, we want to point out that our proposed method is a first attempt to address these issues and that we expect further developments (as stated in the Discussion), for the evaluation of which our dataset could become a useful resource.

Actions taken: We included an additional Supplementary Figure (SFig12) to explore how robust our control-augmentation strategy is with respect to the parameters and data sets chosen and expanded the corresponding discussion (see P14L13-18).

Contribution 4: Discussion and demonstration of some common pitfalls in a microbiome context. As authors note, these are widely known and discussed in stats/ML literature, but there is value in showing them in a subject-specific context as well. I commend the authors on making the effort to both demonstrate these issues with microbiome data, and to make the demonstrations accessible for others to replicate on their own.

We would like to thank the reviewer for the positive assessment of this part of the manuscript.

More minor itemized comments:

P5L27-30: It's important to also discuss metagenomics bias here. See for example recent Willis lab papers (e.g. <https://doi.org/10.1101/761486> and <https://doi.org/10.7554/eLife.46923.001>).

We entirely agree and thank the reviewer for pointing out this oversight.

Actions taken: We added experimental bias and a citation to McLaren et al. to the list of challenges for microbiome data analysis in the revised manuscript at P3L23.

P5L43-44: I agree that (as far as I know) there is no microbiome-specific ML package, and that given the increased usage of such methods in microbiome research that could be very useful.

P5L1-3: Agree completely. This is a huge problem.

We are encouraged to hear that the reviewer agrees with us on the relevance of this issue.

P6L26-30: These are very generic inputs, that seem to place SIAMCAT in the realm of ML methods generally. There doesn't seem to be any utilization of microbiome-specific features such as the hierarchical taxonomy or phylogeny data types.

The reviewer raised an interesting point here. Indeed, we are aware and have in our research group actively investigated methods that take hierarchical dependencies between features into account (for example StructFDR (Xiao et al, PMID: 28505251) or taxonomy-aware feature engineering (Oudah and Henschel, PMID:29907097)). We have however found that objectively evaluating their performance in comparison to classical methods on real data is challenging. Moreover, in many cases we found these more complex methods to be less straightforward to apply (e.g. due to more complex (hyper-)parameter tuning requirements). We therefore feel that substantial additional work is needed to clearly define their benefits before we would want to offer them to users, among whom we specifically also target non-experts. We will continue to investigate these issues and study the work of others for practical recommendations and benchmarks.

Actions taken: We added a section to the discussion about further development and community engagement plans, as mentioned also above in the first area of contribution/criticism.

P6L41-43: This sort of functionality seems available in a number of other places (e.g. metagenomeSeq, or DESeq2).

The reviewer is correct in that there are dedicated packages for testing and visualization of differential abundance testing, some of which are also specific for microbiome data. While this is not the core functionality of SIAMCAT, it often proved useful to compare statistical test results to the features included in machine learning models. Within SIAMCAT, we based the differential abundance testing on the Wilcoxon test (as similarly implemented in LEfSe, Segata et al., PMID: 21702898), since this test performed quite well in a recent systematic comparisons of differential abundance testing methods in microbiome data that notably found highly elevated false discovery rates for many other methods, including metagenomeSeq or edgeR (Hawinkel et al., PMID:28968702).

Actions taken: We included a citation to Hawinkel et al. and the motivation for using the Wilcoxon test on P5L20-23.

P6L46: What does SIAMCAT add relative to other dedicated ML packagers (caret is one, but there are many others)?

As this relates to the first area of contribution/criticism, we hope that this question has been answered by our response above.

P9L7-9: This is a very nice proof-of-principle on this issues. Question: Can SIAMCAT models include confounders like this?

We thank the reviewer for their appreciation of the described issue. SIAMCAT can add meta-variables to the feature matrix before model fitting through the `add.meta.pred` function. This has been used to explore whether a meta-variable might hold orthogonal information compared to the microbiome-based classification, e.g. for combination tests with improved model accuracy (see for example the combination of microbiome features and the FOBT test for the detection of CRC, Zeller et al. PMID:25432777). More complicated comparisons of models with and without possible confounders (for example, likelihood ratio tests), are not yet implemented in the package. However, we plan to extend the package in this direction in the near future, as soon as it is a bit clearer to us (and others in the field) which of these epidemiological concepts are applicable to microbiome data and whether additional customisation is required (for zero-inflated, non-normal, etc. microbiome data).

P10L34-35: This is useful and has value as a subject-specific demonstration of these issues. But this isn't really about SIAMCAT the software package.

We thank the reviewer for their positive assessment regarding our demonstration of common ML pitfalls. We are fully aware that they are not specific to the SIAMCAT package and mention that they are well-described in the statistics literature. However, having often observed exactly these issues in the microbiome literature made us doubt that they are widely known in the community. As we want to increase the uptake of ML in the microbiome field, we feel obliged to also highlight these issues with domainspecific examples to work against further proliferation of statistically flawed ML model evaluations.

P11L58-59: This is an example of where it's unfortunate the code is not more well integrated with the text.

We thank the reviewer for the suggestions to better integrate the code with the manuscript. Actions taken: In P10L14-15, we included a reference to the new SIAMCAT vignettes.

P12L57: Should say right away (given previous text) that the rest of this section is all discussing intra-study results.

We thank the reviewer for pointing out this issue of phrasing.

Actions taken: We adjusted the introduction to this paragraph on P11L1.

P12L59: Is AUROC>0.75 equivalent to "accurately distinguish"? Not really.

We agree that "accurately distinguish" is worded too strongly in this context.

Actions taken: We adjusted the sentence on P11L2-3.

P13L5-6;L14: "accurate microbiome signatures" are not shown here.

Actions taken: We rephrased the statement by qualifying it on P11L16-17.

P13L17-18: "offers implementations..." Is SIAMCAT implementing ML algorithms? I don't think so, and this is the point to make that clear and also mention the implementations the package is relying on.

We thank the reviewer for having spotted this incorrect phrasing.

Actions taken: We edited the sentence to better reflect that SIAMCAT interfaces with ML algorithms through the mlr package, see P11L18-19.

P13L32-33: We see the same with the RF classifier being good in the sense of being robust to microbiome normalization!

We are encouraged and reassured that these results seem to be reproducible.

P13L42-43: Language is a bit strong for me. E.g. "Comprehensive" is too much IMO given the evaluations conducted.

Actions taken: We rephrased the mentioned sentence on P11L36-38.

P14L46-47: Probably an overestimate, no, given the systematic issues introduced above?

We are not entirely sure what exactly the reviewer refers to. If "by systematic issues introduces above" the reviewer means the ML pitfalls presented in Figure 3, we can assure that we a) removed repeated samples for the same individual and b) folded the feature-selection procedures into the cross-validation for the meta-analysis results presented in Figure 4.

P15L11-12: This is a very interesting question.

We would like to thank the reviewer for their positive comment.

P15L15-16: huh? P15L15-16: Why? Why isn't cross-study predictivity being evaluated? Instead of this... weird choice.

As these two points relate to the second area of contribution/criticism, we hope that they have been resolved by our answer above.

P15L27-28: What about other factors, like population heterogeneity (as was so ably demonstrated above) or the specifics of how cases and controls were defined? There is a huge background in clinical meta-analyses that demonstrates the importance of these sorts of issues that seems to just be ignored here. This may be microbiome focused, but the microbiome is not the extent of the issues one must confront in doing clinically relevant microbiome science!

We would like to thank the reviewer for pointing out the relevance of these factors here.

Actions taken: We extended the paragraph on P13L9-12 to include the issues mentioned above.

P15L44-45: Have to define cross-study "portability" before this. See also previous concerns about this concept.

Fig5c: Is "prediction rate for other diseases" defined yet?

As these two points again relate to the second area of contribution/criticism, we hope that they have been resolved by our answer above.

P17L46-48: Is this still kind of bad, compared to other biomarkers in clinical usage?

We agree with the reviewer that a comparison between microbiome-based biomarkers and those in clinical use would be extremely interesting. However, data on clinical biomarkers is generally not available for most microbiome studies. We therefore feel that this comparison is beyond the scope of our paper. At the very least, SIAMCAT makes these comparisons relatively easy, so we hope it will enable other researchers to perform these analyses in the future.

Actions taken: We mentioned the issue raised by the reviewer in the discussion on P19L35-38.

P20L4-5: Not sure anything here is improving "standardization".

We thank the reviewer for pointing out this phrasing issue.

Actions taken: We changed the sentence on P19L2.

P20L20-21: If this is a software paper, this information needs to be way earlier. To be honest, this was a hard to shake discomfort with this paper... so much of what is was doing was useful and needed in the microbiome field, but it didn't seem to completely acknowledge what is was built on (e.g. mlr, but more than that).

We agree with the reviewer that it would be good to acknowledge the dependencies of the SIAMCAT R package earlier in the manuscript.

Actions taken: We adjusted the first paragraph of the results section to better reflect that SIAMCAT depends on previous developments in the statistical computing and microbiome fields and added reference to the existing R/Bioconductor infrastructure whenever applicable.

P20:L59: "disease-specificity". See also above, but... by who I care? If I predict T2D with 100% accuracy, why do I care how that same model does in predicting obesity?

As this relates to the second area of contribution/criticism of the reviewer, we hope that this point has been sufficiently addressed in our response above.

Second round of review

Reviewer 1

Dr Zeller has addressed my concerns, good work!

Reviewer 2

The authors have addressed all my comments related to the software and ML pitfalls in microbiome science, and the revisions they have made to those parts have improved the manuscript and probably warrant publication on their own merits. However, I remain unconvinced of the way they are evaluating ML methods across studies, and in their control augmentation strategy.

The normal way to evaluate the cross-study transfer of an ML method is to apply the model learned in one study to the data from another study. However, in this paper, the authors have made large use of “cross-study portability” and “prediction rate” metrics that are not what people will expect, are still confusing even on re-reading, and have unclear validity.

Figure S9 is critical (albeit dense and hard to follow) in discerning the author's novel ML evaluation strategy. My understanding is that the authors are forming synthetic datasets of the controls from the test dataset and cases from another dataset (and vice versa) to evaluate cross-study portability (and prediction rate). These synthetic evaluation datasets are pasted together from the studies by using knowledge of the labels. Besides an issue of basic validity (test data should not be synthetically constructed by taking account of the label information in this way), it leads to results that are so difficult to interpret that they are at best meaningless or at worst highly misleading. For example, Fig S9D shows a “prediction rate” from Wen to Kushugolova of 0.92 at a 10% FPR. However, that FPR is calculated from the combination of the Wen controls and the Kushu cases. In fact, the FPR of the Wen-trained model on Kushu is about 50%, as essentially all samples (control and case) in Kushu are predicted as cases. In sum, how are these results on pasted datasets relevant to any real application of ML methods? Such a situation will never arise in a real case where new samples are being collected and then evaluated using a common protocol (i.e. within a common study) and without the labels being known beforehand.

This feeds into concerns about the control augmentation strategy. First, the justification for the strategy is still not convincing, controls are not the same study-to-study and insufficient care is taken here to describe and rationalize the way in which an augmented set of controls that are truly different from the cases in a focal study should be constructed. Second, control augmentation results in a more diverse collection of control samples and therefore leads to a larger proportion of the entire sample space being classified as “control” and fewer things being classified as “cases”. This provides an alternative explanation of the presented improvements from this method: This is a disadvantage when looking at true positive rates for cross-study prediction on the same disease (because more cases are classified as controls) but an advantage when looking at the detection rates for cross-study prediction on different diseases (because false positives are reduced). Third, there is a lingering concern that there is a circularity in the way out-of-study control samples are being added to train the model, and then a mix of training control samples and other-study case samples are being used to test the model. Finally, in comparison of Fig S11 and Fig 5D, it appears that naive ML models (i.e. without control augmentation) actually do on average *better* in the cross-study prediction accuracy within the same disease -- a very important point that is arguably more relevant to any application of these models than the cross-disease performance!

In sum, this has been a difficult paper to evaluate, because it has tremendous strengths already and probably deserves publication as a software and ML pitfalls paper alone. However, the novel ML evaluations and methods sections remains problematic.

Reviewer reports:

Reviewer #1: Dr Zeller has addressed my concerns, good work!

We are grateful to this reviewer for their positive assessment of our revised manuscript.

Reviewer #2: The authors have addressed all my comments related to the software and ML pitfalls in microbiome science, and the revisions they have made to those parts have improved the manuscript and probably warrant publication on their own merits. However, I remain unconvinced of the way they are evaluating ML methods across studies, and in their control augmentation strategy.

The normal way to evaluate the cross-study transfer of an ML method is to apply the model learned in one study to the data from another study. However, in this paper, the authors have made large use of “cross-study portability” and “prediction rate” metrics that are not what people will expect, are still confusing even on re-reading, and have unclear validity. *We agree with the reviewer that the measures we employ in our meta-analysis are not the standard measures used in transfer of ML models across studies of the same disease. We would like to apologize for a lack of clarity in motivating why we feel that these are useful. We would also like to emphasize that these new measures enable analyses beyond classical transfer of models between studies and are not meant to replace these. As detailed below, we have now included a more in-depth presentation of the classical model transfer across studies of the same disease using standard evaluations and added additional motivation/explanation for their extension with the newly introduced evaluation measures which enable comparisons across studies for different diseases.*

Figure S9 is critical (albeit dense and hard to follow) in discerning the author's novel ML evaluation strategy. My understanding is that the authors are forming synthetic datasets of the controls from the test dataset and cases from another dataset (and vice versa) to evaluate cross-study portability (and prediction rate). These synthetic evaluation datasets are pasted together from the studies by using knowledge of the labels. Besides an issue of basic validity (test data should not be synthetically constructed by taking account of the label information in this way), it leads to results that are so difficult to interpret that they are at best meaningless or at worst highly misleading. For example, Fig S9D shows a “prediction rate” from Wen to Kushugolova of 0.92 at a 10% FPR. However, that FPR is calculated from the combination of the Wen controls and the Kushu cases. In fact, the FPR of the Wen-trained model on Kushu is about 50%, as essentially all samples (control and case) in Kushu are predicted as cases. In sum, how are these results on pasted datasets relevant to any real application of ML methods? Such a situation will never arise in a real case where new samples are being collected and then evaluated using a common protocol (i.e. within a common study) and without the labels being known beforehand.

In the “standard” case of cross-study transfer of ML models (to which we believe the reviewer refers), one would usually deal with datasets of the same disease and could therefore evaluate the ML model transfer in a straightforward way using ROC analysis, as we have done for example in a recent colorectal cancer meta-analysis (Wirbel et al. 2019, PMID: 30936547). In this case, both the false-positive rate (FPR) and the true-positive rate (TPR), which are combined in a ROC plot, can be calculated on the external test dataset. We have now presented this in a new SFig9 in detail and agree that this will be familiar to most readers.

We are however unsure, what the reviewer means in their discussion of the Wen / Kushugolova example. In particular, we would like to remark that the FPR is (always) calculated exclusively on control examples, the TPR exclusively on cases. As a consequence, it is also straightforward to evaluate the false-positive rate -- but not the true-positive rate -- on any external data set that does not contain cases of the disease the model was trained to recognize. Such a straight-forward analysis of external FPRs is now also included in Fig S9.

In our view, which perhaps is not in agreement with the reviewer's, evaluations going beyond the few microbiome datasets that are currently available for each disease are very relevant -- often even necessary -- for a thorough assessment of the robustness of ML models to study-to-study heterogeneity. This is because tight control of the FPR is crucial in essentially all biomedical applications and the relatively small size of most existing microbiome studies exacerbates the need for cross-study evaluations.

When evaluating ML models for a given disease on datasets from a different disease, we are convinced that it is meaningful to consider the outcome of this evaluation separately for the cases and controls in the external test set. This is clear from prior work in the field (Duvall et al. 2017, PMID: 29209090, Pasolli et al. 2016, PMID: 27400279) and the observation that FPRs can behave very differently for the plausible reason that there may be microbiome changes that are shared across diseases (e.g. due to a shared inflammatory milieu, Duvall et al. 2017, PMID: 29209090), which could lead to cross-recognition; but these are not manifest in the respective control samples. The main concern in this case should however be if an elevated prediction rate (FPR) of a colorectal cancer classifier on a Crohn's disease dataset, say, is due to common biological processes or just a technical artefact -- which does not appear unlikely for naive model transfer given the excessive FPRs we observed (Fig S9). This motivated us to introduce a measure for a classifier's prediction rate on other diseases. This "prediction rate for other diseases" is a straightforward evaluation of its FPR on samples from other diseases (at a prediction score cutoff fixed at a value at which 10% of the original control samples would be positive, thus 10% FPR). We have difficulties to see how this measure could be "misleading" and for the above-stated reason also respectfully disagree with the reviewer's concern that it may be "meaningless". There might have been a misunderstanding in the interpretation of the former Fig S9 in that it may not have been clear how the prediction score cutoff was chosen. We just calculated the percentage of cases in the external dataset that is above a prediction cutoff corresponding to a 10% FPR in cross validation; 92% in the example highlighted by the reviewer. We are not sure how the reviewer would arrive at the 50% number in his/her comment. For clarification we have revised the Supplementary Figures and added more details to the explanation in the manuscript.

When transferring ML models for a given disease to datasets from another disease, the AUC cannot be calculated in the standard way (as TPR cannot be estimated). Consequently, the key advantage of a ROC plot, which is its representation of all possible cutoffs on the decision function corresponding to different trade-offs between FPR and TPR, cannot be well captured by evaluating the FPR for a fixed cutoff. To address this, we introduced one additional measure for evaluating ML models after careful consideration of several alternatives. This measure, which we call "cross-study portability", is analogous to the AUC in that it assesses if the classifier is able to distinguish controls in the external dataset from

the disease cases, albeit from the original dataset (as the external dataset does not contain any). Rather than an evaluation on “synthetic datasets [...] pasted together”, in our view this can be seen as a ROC analysis of an external FPR (estimated on controls of the external dataset for a different disease) against an internal TPR (estimated on cases from the cross-validation set), which we feel is more informative than standard evaluations of FPRs at an arbitrarily chosen cutoff. Of course -- as any other evaluation of classifier performance -- this is based on label information as a ground truth. However, there is no way for the label information of the external dataset to leak into the training of the ML model, since training is done through internal cross-validation on the original dataset. We therefore don't share the reviewer's concern about basic statistical validity.

In summary, we hope that these clarifications have clarified remaining reservations about the validity and relevance of our evaluation measures. To address the concern that they might be “misleading”, we have added more basic evaluations -- which, within their more limited scope of application, do support our original conclusions (Fig S9). The relevance of these evaluation measures to the real-world application of ML models had already been explained in the Discussion on page 13 of the manuscript following the previous suggestions of this reviewer, which we would like to reiterate here due to its importance. An ML model trained to detect a rare disease (such as cancer) should not show elevated predictions on a common disease (such as T2D), otherwise the model is unusable in a real-world diagnostic setting due to a lack of precision (among its predictions will be many more diabetes than cancer patients). Additionally, we and other researchers previously working on this topic (e.g. Duvall et al. 2017, PMID: 29209090, Pasolli et al. 2016, PMID: 27400279), do believe that disease-specificity of microbiome signatures is of interest also for its biological implications. Tools to properly assess it in the face of excessive technical heterogeneity across microbiome data sets are thus urgently needed.

Actions taken:

Recognizing that some confusion on our evaluation of disease specificity and cross-study portability still remained, we (i) included the standard within-disease cross-study evaluations the reviewer referred to (Fig S9), (ii) included another paragraph in the main text (page 14) to motivate why the newly introduced measures go beyond these and what they are useful for. Additionally, we (iii) tried to make Figure S10 (formerly Fig S9) easier to understand by clarifying how both measures are computed and highlighting how the 10% FPR cutoff is derived.

This feeds into concerns about the control augmentation strategy. First, the justification for the strategy is still not convincing, controls are not the same study-to-study and insufficient care is taken here to describe and rationalize the way in which an augmented set of controls that are truly different from the cases in a focal study should be constructed.

We welcome the critical assessment of our control-augmentation strategy by Reviewer #2. As we have stated in the figure legend for the former Figure S12 already, we fully acknowledge that “control” is not a clear concept and definitions can vary considerably between studies. However, at an operational level it is a useful approach to enrich for asymptomatic individuals to reduce the likelihood that they suffer from the disease they are compared to, even though this cannot be ruled out completely. Especially from the standpoint of post-publication data analysis, there is basically no alternative to using each study's “control” designation as is. The reviewer's theoretical concern can essentially be

brought forward against any meta-analysis, and while it clearly is a limitation, it does not render meta-analyses invalid in general.

Actions taken:

Since our discussion on the topic within the figure legend of Figure S12 seemed insufficient, we now included statements in the main text that both cross-study evaluations and control-augmentation rely on, and are limited by, their assumption on controls being comparable between studies (pages 13 and 14).

Second, control augmentation results in a more diverse collection of control samples and therefore leads to a larger proportion of the entire sample space being classified as "control" and fewer things being classified as "cases". This provides an alternative explanation of the presented improvements from this method: This is a disadvantage when looking at true positive rates for cross-study prediction on the same disease (because more cases are classified as controls) but an advantage when looking at the detection rates for cross-study prediction on different diseases (because false positives are reduced).

This point raised by the reviewer is interesting and the concern plausible. To ascertain whether our approach leads to a reduced true-positive rate, we checked ML model transfer accuracy for datasets of the same disease before and after control-augmentation (Fig S9a). This comparison in fact showed that control-augmented models generally show very similar or even higher AUC values than naive models (in cross-study evaluations within a disease), which means that despite the broadening of the control-space, the disease signal is still well discernible for the ML algorithm.

Actions taken:

We included the AUROC evaluation on external datasets for the same disease before and after control-augmentation into the new Fig S9a.

Third, there is a lingering concern that there is a circularity in the way out-of-study control samples are being added to train the model, and then a mix of training control samples and other-study case samples are being used to test the model.

This concern is not justified, as control samples are added from entirely independent data sets during model fitting (independently to each of the training sets used during cross-validation), while the test sets remain unchanged. The cohort studies we used for augmentation were solely used for this purpose and not included in any of the evaluations. Therefore, leakage of label information across cross-validation folds could not have occurred. For the additional robustness analysis carried out upon the reviewer's earlier request, in which controls from datasets from within the ML meta-analysis were used for augmentation (former Fig S12), the augmented models are then not evaluated on these datasets, precisely to prevent leakage of testing information into the training set and circular reasoning.

Finally, in comparison of Fig S11 and Fig 5D, it appears that naive ML models (i.e. without control augmentation) actually do on average better in the cross-study prediction accuracy within the same disease -- a very important point that is arguably more relevant to any application of these models than the cross-disease performance!

The concern that the naive models without control-augmentation might on average do better when looking at the prediction rate for the same disease, is valid and related to the second

point raised by this reviewer. We had already noted this observation in the figure caption of Fig S13 (former Fig S12) showing the difference between control-augmented and naive models for a **single tradeoff** between disease specificity and true-positive rate for the same disease. However, the newly added Fig S9 clearly shows that across the range of possible cutoffs (assessed in a ROC curve), **control-augmented models perform equally or better** than naive models also in transfer across studies of the same disease. We therefore conclude that control-augmentation does **not** reduce cross-study prediction accuracy within the same disease.

In sum, this has been a difficult paper to evaluate, because it has tremendous strengths already and probably deserves publication as a software and ML pitfalls paper alone.

However, the novel ML evaluations and methods sections remains problematic.

We would like to thank the reviewer for their positive assessment of the first parts of our manuscript. We hope that our additional analysis and evaluation now convinced them that the control-augmentation approach and cross-study evaluations hold great potential for advancing ML-based microbiome meta-analysis. In particular, by showing them side-by-side with more standard evaluations into the revised manuscript, we hope that it becomes clear that they were never meant to mislead and add meaningful information to the manuscript as a whole. In our eyes, fortifying ML models against some of the technical heterogeneity among microbiome studies to enable biologically plausible cross-study application is an important contribution to the ongoing debate about broad (e.g. inflammation-related) versus specific microbiome signatures, the latter of which are clearly needed for future clinical translation.